# BAG OF TRICKS TO BOOST ADVERSARIAL TRANSFER-ABILITY

## ABSTRACT

Deep neural networks are widely known to be vulnerable to adversarial examples. However, vanilla adversarial examples generated under the white-box setting often exhibit low transferability across different models. Since adversarial transferability poses more severe threats to practical applications, various approaches have been proposed for better transferability, including gradient-based, input transformation-based, and model-related attacks, *etc*. In this work, we find that several tiny changes in the existing adversarial attacks can significantly affect the attack performance, *e.g.*, the number of iterations and step size. Based on careful studies of existing adversarial attacks, we propose a bag of tricks to enhance adversarial transferability, including momentum initialization, scheduled step size, dual example, spectral-based input transformation, and several ensemble strategies. Extensive experiments on the ImageNet dataset validate the high effectiveness of our proposed tricks and show that combining them can further boost adversarial transferability. Our work provides practical insights and techniques to enhance adversarial transferability, and offers guidance to improve the attack performance on the real-world application through simple adjustments.

## 1 INTRODUCTION

Adversarial examples, in which imperceptible perturbations to benign samples can deceive deep neural networks (DNNs), have attracted significant attention in recent years (Szegedy et al., 2013; Goodfellow et al., 2015; Agarwal et al., 2022). These examples highlight the vulnerability of DNNs and raise critical security concerns in various domains, including autonomous driving (Cao et al., 2019; Nesti et al., 2022; Girdhar et al., 2023), facial authentication (Chen et al., 2017; 2019; Joos et al., 2022), object detection (Li et al., 2021; Nezami et al., 2021; Zhang & Wang, 2019), *etc*. The study of adversarial examples has spurred researches on enhancing the robustness (Madry et al., 2018; Shafahi et al., 2019; Zheng et al., 2020; Jia et al., 2022) and understanding (Shumailov et al., 2019) of DNNs. Overall, adversarial examples have become a crucial tool for uncovering vulnerabilities and boosting the robustness of DNNs.

Without any knowledge (*e.g.*, architecture, parameters, training loss functions, *etc*.) about remote victim models used in real-world applications, attackers usually employ the local surrogate model to craft adversarial examples to deceive the victim models, dubbed adversarial transferability. To improve the attack success rate against the victim model using the surrogate model, numerous approaches have been proposed to improve the adversarial transferability, such as gradient-based attacks (Dong et al., 2018; Wang & He, 2021; Wang et al., 2021b), input transformation-based attacks (Xie et al., 2019; Dong et al., 2019; Wang et al., 2021a), model-related attacks (Liu et al., 2017; Xiong et al., 2022; Gubri et al., 2022), *etc*.

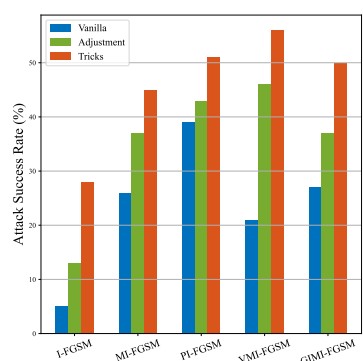

Figure 1: Attack success rates (%) of 100 adversarial examples against Google's vision API with VGG-16 as the surrogate model. We denote Vanilla as the general setting, Adjustment as adjusting the hyper-parameters, and Tricks as being integrated with our tricks.

Existing transfer-based attacks generally follow a unified setting, which sets the number of iterations $T$ as 10, the momentum decay factor $\gamma$ as 1, the step size $\alpha$ as $\frac{1}{T}$, *etc*. However, it does not represent the actual performance in real-world applications. As shown in Fig. 1, when employing VGG-16 to generate 100 adversarial examples to attack Google's vision API [1] using I-FGSM (Goodfellow et al., 2015), MI-FGSM (Dong et al., 2018), PI-FGSM (Gao et al., 2020), VMI-FGSM (Wang & He, 2021), and GIMI-FGSM (Wang et al., 2022a), respectively, the average attack success rate over different methods under the general setup (Vanilla) is 23.6%, while it has a large improvement of 11.6% by the hyper-parameters adjustment (Adjustment, see Appendix A). It supports that simple adjustment has a great impact on adversarial transferability, leading to performance differences in applications. This finding motivates us to investigate the real performance of different attacks.

To give a comprehensive study, we first investigate the impact of key hyper-parameters on adversarial transferability in generally used gradient-based attacks, which also serve as the backbone of other methods. Then, we propose a collection of simple yet effective tricks to boost the performance of different categories of attacks, including 1) gradient-based attacks: random initialization, scheduled step size, dual example; 2) input transformation-based attacks: high-frequency transformation, spectral-based augmentations; 3) ensemble-based attacks: gradient alignment, asynchronous input transformation, and model shuffle. As shown in Fig. 1, the integrated tricks (Tricks) further bring 10.8% improvement of the mean attack success rate against Google's vision API. Extensive experiments on the ImageNet dataset demonstrate that our tricks help overcome the limitations and challenges in crafting adversarial examples and significantly improve the attack performance of the selected dozen attacks. By combining these tricks, we can achieve superior attack performance on defense models and Google's vision API, which reveals the inefficiency of existing defense mechanisms and scalability in real-word applications.

## 2 RELATED WORK

### 2.1 ADVERSARIAL ATTACK AND ADVERSARIAL TRANSFERABILITY

Since Szegedy et al. (2013) uncovered the vulnerability of DNNs to adversarial examples, numerous adversarial attacks have been proposed, including 1) *white-box attacks*: the attacker has the full knowledge of the victim model (Goodfellow et al., 2015; Moosavi-Dezfooli et al., 2016; Carlini & Wagner, 2017), *e.g*., architecture, loss function, *etc*. 2) *black-box attacks*: the attacker has no prior information of the victim model. It is often impossible to access information about the target victim model in real-world scenarios, necessitating black-box attack techniques. Existing black-box attacks can be grouped into three classes: score-based (Andriushchenko et al., 2020; Yatsura et al., 2021), decision-based (Li et al., 2022; Chen et al., 2020; Wang et al., 2022b), and transfer-based (Dong et al., 2018; Lin et al., 2020; Wang et al., 2021a) attacks. Score-based and decision-based attacks typically require a significant number of queries on the victim model, while transfer-based attacks adopt the adversarial examples generated on surrogate models to fool different victim models. This makes transfer-based attacks more computationally efficient and better suited for real-world applications. Hence, we focus on transfer-based attacks. Numerous researchers have devised strategies to enhance adversarial transferability, concentrating mainly on three approaches: iterative gradient-based optimization, input transformation-based methods, and model-related techniques.

**Gradient-based optimization methods**. I-FGSM (Kurakin et al., 2018) extends FGSM (Goodfellow et al., 2015) into an iterative version to substantially enhance the attack effectiveness under the white-box setting but exhibit poor transferability. MI-FGSM (Dong et al., 2018) incorporates momentum to improve adversarial transferability, while NI-FGSM (Lin et al., 2020) applies Nesterov momentum for optimization acceleration. PI-FGSM (Gao et al., 2020) recycles the clipped adversarial perturbation to the neighbor pixels to enhance the transferability. VMI-FGSM (Wang & He, 2021) adjusts the gradient based on the gradient variance of the previous iteration to stabilize the update direction. EMI-FGSM (Wang et al., 2021b) enhances the momentum by averaging the gradient of data points sampled from the optimization direction. GIMI-FGSM (Wang et al., 2022a) initializes the momentum by running the attacks in several iterations for gradient pre-convergence.

**Input transformation methods**. Input transformation-based attacks have shown great effectiveness in improving transferability and can be combined with gradient-based attacks. For instance, diverse

---

[1]https://cloud.google.com/vision

input method (DIM) (Xie et al., 2019) resizes the input image to a random size, which is then padded to a fixed size for gradient calculation. TIM (Dong et al., 2019) adopts Gaussian smooth on the gradient to approximate the average gradient of a set of translated images to update the adversary. Scale-invariant method (SIM) (Lin et al., 2020) calculates the gradient on a collection of scaled images. *Admix* (Wang et al., 2021a) incorporates a fraction of images from other categories into the inputs to generate multiple images for gradient calculation. SSA (Long et al., 2022) randomly transforms the image in the frequency domain to craft more transferable adversarial examples.

**Model-related methods**. Liu et al. (2017) initially discovered that an ensemble attack, which generates adversarial examples on multiple models, can result in better transferability. Li et al. (2020) simultaneously attack several ghost networks, which are generated by adding dropout layers to the surrogate model. Xiong et al. (2022) minimize the gradient variance across different models to enhance ensemble attacks. Gubri et al. (2022) train the model with a high learning rate to produce multiple models and attack them sequentially to improve existing attacks' transferability.

## 2.2 ADVERSARIAL DEFENSE

To mitigate the threat of adversarial attacks, a variety of defense methods have been proposed, *eg.*, adversarial training (Goodfellow et al., 2015; Zhang et al., 2019; Wang et al., 2020a), input pre-processing (Guo et al., 2018), certified defense (Cohen et al., 2019) *etc*. For example, Liao et al. (2018) propose a high-level representation guided denoiser (HGD) to purify the adversarial examples. Madry et al. (2018) introduce an adversarial training method (AT) that utilizes PGD adversarial examples to train models, aiming to enhance their adversarial robustness. Wong et al. (2020) employ random initialization in FGSM adversarial training, leading to Fast Adversarial Training (FAT), which achieves accelerated training and improved adversarial robustness comparable to PGD training. Cohen et al. (2019) propose a random smoothing technique (RS) to provide the model with certified robustness against the adversarial examples. Naseer et al. (2020) design a neural representation purifier (NRP) to remove harmful perturbations of images to defend adversarial attacks.

## 3 BAG OF TRICKS

In this study, we explore various techniques to generate more transferable adversarial examples. We evaluate the effectiveness of these factors and tricks using several attacks, including FGSM (Goodfellow et al., 2015), I-FGSM (Kurakin et al., 2018), MI-FGSM (Dong et al., 2018), NI-FGSM (Lin et al., 2020), VMI-FGSM (Wang & He, 2021), EMI-FGSM (Wang et al., 2021b), and GIMI-FGSM (Wang et al., 2022a). In our default setting, we choose $10,000$ images from the ImageNet-1K dataset as our evaluation set. We use seven surrogate/victim models, comprising 1) Convolutional Neural Network (CNNs): VGG-16 (Simonyan & Zisserman, 2015), ResNet-18 (He et al., 2016), ResNet-101 (He et al., 2016), DenseNet-121 (Huang et al., 2017), and MobileNet (Howard et al., 2017); and 2) transformers: Vit (Dosovitskiy et al., 2020) and Swin (Liu et al., 2021). We generate adversarial examples using VGG-16 and evaluate their performance on the other seven models by reporting the mean attack success rate. To align with previous works, we set the maximum perturbation magnitude $\epsilon = \frac{16}{255}$. Unless otherwise specified (denoted as Ori.), we set the number of iterations as 10, the step size as $\frac{1.6}{255}$, momentum decay factor $\gamma$ as 1, the look-ahead factor for NI-FGSM as $\frac{1.6}{255}$, the number of additional samples used in EMI-FGSM and VMI-FGSM as 11 and 20, the number of pre-computing epochs for GIMI-FGSM as 5. We give detailed experiment settings in Appendix A and more results using different models as the surrogate model in Appendix E and F.

## 3.1 HYPER-PARAMETERS STUDY

When fixing the maximum perturbation and evaluation criteria, the other hyper-parameters, such as the number of iterations, step size, and momentum coefficient, might greatly impact performance but be overlooked in existing works. Here, we conduct experiments to examine the effects of the number of iterations $T$, the scale factor $s$ for step size, and the momentum decay factor $\gamma$ on performance.

**On the number of iterations** $T$: After MI-FGSM adopts $T = 10$ for evaluations, the subsequent works take $T = 10$ as the default setting. However, it remains unexplored how varying the value of $T$ impacts different attack methods. Here we conduct a series of experiments with $T$ ranging from 1 to 20 to analyze its influence on various attack techniques. As shown in Fig. 2a, increasing $T$

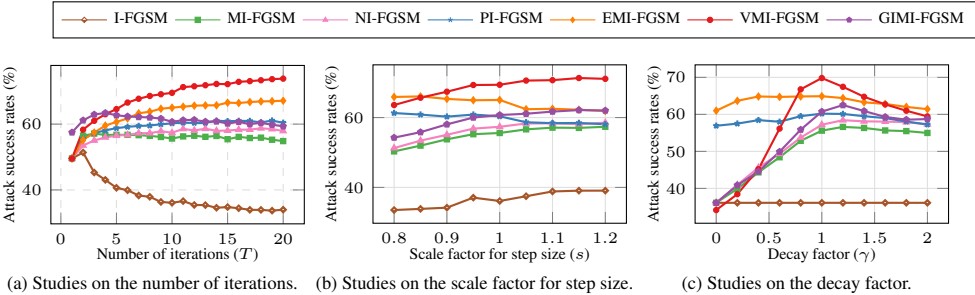

(a) Studies on the number of iterations. (b) Studies on the scale factor for step size. (c) Studies on the decay factor.

Figure 2: Hyper-parameters studies of seven iterative adversarial attacks on the number of iterations, scale factor for step size and decay factor.

significantly affects the performance of I-FGSM, indicating the crucial role of near-sample gradients in boosting the adversarial transferability. Conversely, for the other baseline methods, augmenting the number of iterations initially improves their attack performance. GIMI-FGSM, in particular, achieves its peak performance at $T = 5$, showing the effectiveness of the global momentum in skipping the local optima near samples but still suffering from performance degradation for larger values T due to multiple local optimas far away from the samples. Notably, VMI-FGSM consistently improves performance as the number of iterations increases. This suggests that the variance tuning mechanism of VMI-FGSM effectively stabilizes the update direction, enabling it to escape poor local minima and achieve superior results. Hence, the optimal number of iterations varies for various attacks and it is crucial to set the appropriate value of $T$ for superior performance.

**On the scale factor $s$ for step size**: It is common to set the step size $\alpha = \epsilon/T$ for generating adversarial examples. However, recent researches (Gao et al., 2020; Xie et al., 2021) have indicated that adopting a larger step size can enhance the transferability of adversarial examples. To investigate this further, we conduct experiments by scaling the step size with a factor ranging from $0.8\times$ to $1.2\times$ relative to $\epsilon/T$. The results are illustrated in Fig. 2b. We can see that increasing the step size yields improved performance for several momentum-based attack methods, indicating an adequately large step size can be utilized to skip over the local optima and boost the adversarial transferability. However, it is important to note that excessively large step sizes can overlook the optima and cause inefficient optimization under $l_\infty$ constraint, thus harming attack performance, especially for PI-FGSM. Therefore, it is crucial to carefully consider an adequately large step size that balances the improved transferability without performance degradation.

**On the momentum decay factor $\gamma$**: Typically, the decay factor for momentum is set to 1, indicating no decay, while $\gamma < 1$ indicates the increasing impact of momentum (Wang et al., 2021b). To investigate the impact of $\gamma$, we conduct experiments using different momentum decay factors, ranging from 0 to 2 with a step size of 0.2. As shown in Fig. 2c, setting $\gamma$ as 1.0 or 1.2 leads to better transferability. A small value of $\gamma$ tends to result in degradation towards the behavior of I-FGSM, while a large value of $\gamma$ leads to the dominance of historical momentum. It is important to set an appropriate momentum decay factor to balance the historical momentum and novel gradients, thus better searching for the direction towards optima to boost the adversarial transferability.

**Takeaways**. In general, the optimal number of iterations and scale factor for step size varies for different attacks. Achieving superior performance necessitates a meticulous process of finetuning. For most attacks, a momentum decay factor $\gamma = 1.0$ or $\gamma = 1.2$ is appropriate. Besides, the results of the hyper-parameter study suggest that adversarial transferability is much more sensitive to some basic settings than we thought. We appeal to more concerns on the overlooked confounders when benchmarking attacks. More discussions are provided in Appendix G.

## 3.2 MOMENTUM INITIALIZATION

Initialization techniques (*e.g*., random start, Xavier (Glorot & Bengio, 2010), Kaiming (He et al., 2015)) have been recognized as crucial methods to expedite convergence in optimization problems. As depicted in Fig. 2c, the momentum takes a crucial role in stabilizing the optimization direction. However, there is limited research on leveraging better momentum initialization techniques to enhance adversarial transferability. Recently, GIMI-FGSM initializes the global momentum by running MI-FGSM for a few iterations, which achieves better adversarial transferability. However, accurately

Table 1: Average attack success rates (%) of momentum-based attacks with GI or RGI for global momentum initialization, including MI-FGSM, NI-FGSM, PI-FGSM, EMI-FGSM, and VMI-FGSM. We report the performance under the default setting ($T = 10$)/optimal setting. We denote "Ori." as the vanilla attacks.

|  | MI-FGSM | NI-FGSM | PI-FGSM | EMI-FGSM | VMI-FGSM |
|---|---|---|---|---|---|
| Ori. | 55.6/56.7 | 57.4/58.6 | 60.2/61.0 | 65.0/67.1 | 69.5/73.8 |
| GI | 60.7/61.7 | 56.9/57.9 | 60.7/62.2 | 68.2/68.2 | 70.6/75.2 |
| RGI | **64.4/64.8** | **64.8/65.1** | **69.6/69.6** | **73.3/73.3** | **78.1/79.3** |

capturing the true global momentum and pre-computing it becomes challenging due to the presence of multiple local optima around the initial benign samples.

To address this issue, we posit that the initialization of the global momentum warrants a thorough examination of the entire surrounding region. Inspired by VMI-FGSM and EMI-FGSM, we randomly sample several samples in the $\epsilon$-neighborhood of input image $x$ to accumulate the momentum as the global momentum, denoted as random global momentum initialization (RGI). By incorporating RGI, we aim to *capture a more representative global momentum that takes into account the diverse local optima surrounding the initial benign sample*. The detailed algorithm is in Appendix B.1.

**Results**. Since GIMI-FGSM initializes the global momentum with 5 iterations, we sample 5 examples in RGI for a fair comparison. Table 1 demonstrates the impact of different initialization methods. It can be observed that Global Initialization (GI) only exhibits a noticeable effect on a few baselines, failing to enhance the adversarial transferability of NI-FGSM and PI-FGSM. In contrast, RGI consistently improves the adversarial transferability for all the baselines, surpassing GI by a remarkable margin. This outcome confirms the effectiveness of momentum initialization and supports our argument that proper initialization of the global momentum necessitates comprehensive exploration of the neighborhood.

### 3.3 SCHEDULED STEP SIZE

Although I-FGSM achieves superior white-box performance, as shown in our hyper-parameter study of Fig. 2a, it exhibits poorer adversarial transferability than FGSM ($T = 1$). Interestingly, I-FGSM exhibits poorer adversarial transferability as the number of iterations increases. This inspires us that *the gradient of data points close to the input image plays a crucial role in generating transferable adversarial examples*. To balance the white-box performance and transferability, we adjust the importance of the gradient at each iteration by using various step sizes to update the perturbation. In particular, we optimize $x^{adv}$ as follows:

$$x_{t+1}^{adv} = \text{Clip}_{[x-\epsilon,x+\epsilon]} \left\{ x_t^{adv} + \epsilon \cdot \hat{\alpha}_t \cdot \text{sign}\left( \nabla_{x_t^{adv}} \mathcal{L}\left( f\left( x_t^{adv} \right), y \right) \right) \right\}, \tag{1}$$

where $\text{Clip}_{[x-\epsilon,x+\epsilon]}(\cdot)$ clips the input into $[x - \epsilon, x + \epsilon]$, $\hat{\alpha}$ is the scheduled increasing step size sequence and $x_0^{adv} = x$.

**Results**. To validate the effectiveness of scheduled step size, we adopt five sequences, namely logarithm: $\alpha_i = \frac{\ln(T-i)}{\sum_{k=1}^{T} \ln(T-k)}$, linear: $\alpha_i = \frac{T-i}{\sum_{k=1}^{T} (T-k)}$, exponential: $\alpha_i = \frac{e^{T-i}}{\sum_{k=1}^{T} e^{T-k}}$, and pvalue: $\alpha_i = \frac{1/i^p}{\sum_{k=1}^{T} 1/k^p}$. These five sequences are ploted in Fig. 3. We can see that they exhibit various

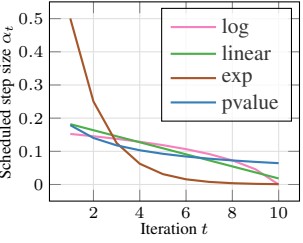

Figure 3: Scheduled step size at various iterations.

Table 2: Average attack success rates (%) of gradient-based attacks using various scheduled step sizes, including I-FGSM, MI-FGSM, NI-FGSM, PI-FGSM, EMI-FGSM, VMI-FGSM, and GIMI-FGSM. We report the performance under the default setting ($T = 10$)/optimal setting. We denote "Ori." as the identity step size.

|  | I-FGSM | MI-FGSM | NI-FGSM | PI-FGSM | EMI-FGSM | VMI-FGSM | GIMI-FGSM |
|---|---|---|---|---|---|---|---|
| Ori. | 36.1/51.6 | 55.6/56.7 | 57.4/58.6 | 60.2/61.0 | 65.0/67.1 | 69.5/73.8 | 60.7/61.7 |
| log | 35.9/49.5 | 56.9/57.1 | 58.6/59.1 | 60.8/61.2 | 65.9/67.4 | **73.9/74.1** | 61.1/61.9 |
| linear | 35.7/46.2 | 56.4/56.9 | **60.4/60.7** | **61.2/62.0** | **66.7/68.3** | 70.3/70.9 | 60.9/60.9 |
| exp | 37.8/46.2 | **59.9/59.9** | 57.9/57.9 | 52.1/52.1 | 63.8/64.5 | 71.1/71.1 | 58.8/59.0 |
| pvalue | **38.4/52.2** | 57.7/57.7 | 58.3/59.0 | 56.1/56.2 | 64.2/64.9 | 69.7/70.5 | **61.8/62.5** |

Table 3: Average attack success rates (%) when applying dual example w/o or w ensemble strategy using various sequences as step size to various attacks, *i.e.*, I-FGSM, MI-FGSM, NI-FGSM, PI-FGSM, EMI-FGSM, VMI-FGSM, and GIMI-FGSM. We denote "N/A" as vanilla adversarial attacks, and others as dual example with scheduled step size.

|        | I-FGSM    | MI-FGSM   | NI-FGSM   | PI-FGSM   | EMI-FGSM  | VMI-FGSM  | GIMI-FGSM |
|--------|-----------|-----------|-----------|-----------|-----------|-----------|-----------|
| N/A    | 36.1      | 55.6      | 57.4      | 60.2      | 65.0      | 69.5      | 60.7      |
| Ori.   | 48.9/62.3 | 56.1/65.5 | 59.3/64.0 | 62.8/66.6 | 66.3/68.9 | 75.2/77.3 | 63.4/71.0 |
| log    | **50.6**/**62.9** | 60.7/**66.9** | **62.5**/66.9 | 56.6/66.0 | 63.8/**69.4** | **77.9**/78.7 | 66.8/**71.2** |
| linear | 40.9/62.0 | 60.9/66.6 | 61.7/**67.3** | **63.5**/66.9 | **67.1**/**69.4** | 76.9/**78.8** | **68.1**/70.6 |
| exp    | 39.3/61.9 | **61.5**/65.6 | 61.3/65.8 | 60.6/66.4 | 65.9/67.3 | 70.5/74.9 | 63.4/69.1 |
| pvalue | 48.9/62.1 | 57.7/66.1 | 62.0/64.2 | 62.8/**67.0** | 66.2/68.7 | 75.6/77.9 | 66.6/69.9 |

decreasing rates, resulting in different magnitudes of significance for each iteration. In Tab. 2, we present the results of applying the scheduled step size to seven attacks. We can observe that the scheduled step size consistently improves the adversarial transferability of all seven attacks. For example, the logarithmic sequence yields a significant improvement of $4.4\%$ for VMI-FGSM, indicating the considerable potential of the scheduled step size approach in enhancing transferability by carefully adjusting the step size. However, it is worth noting that the optimal scheduled sequence for each attack method differs from one another. This highlights the importance of tailoring the step size schedule to the specific characteristics and requirements of each attack. These findings highlight that researchers can *enhance adversarial transferability for free by carefully considering the decline rates of different sequences and their impact on the importance of each iteration.*

### 3.4 DUAL EXAMPLE

Although the scheduled step size can boost the adversarial transferability, it might significantly diminish the gradient of the latter iterations. This is especially evident when the step size decreases significantly over time. For instance, using an exponential sequence that yields the best performance on MI-FGSM, the step size becomes close to zero after $t = 5$. Consequently, the last five iterations have a negligible effect on updating the adversarial perturbation. To address this issue, we aim to *strike a balance between utilizing gradients near the benign samples for enhanced transferability and achieving efficient updates on the adversarial perturbations during the later steps.* By doing so, we aim to overcome the limitations imposed by a diminishing step size and ensure that the later iterations contribute meaningfully to the attack process.

The core concept behind using a scheduled step size is to prioritize the gradients of data points that are closer to the input image. Scheduled step size gives more weight to the gradients of data points close to the input image by assigning large step sizes in the first several iterations. In addition to using a scheduled step size, an alternative approach is to adopt the gradient of data points close to the input image more frequently. This means that during the iterative attack process, the gradients of nearby data points are utilized multiple times, thereby placing greater importance.

Hence, we propose a new strategy that builds upon existing attacks but incorporates the use of gradients calculated on a dual example. The dual example is optimized by I-FGSM with an *increasing* sequence as the step size, resulting in a larger number of data points for gradient calculation. To further emulate the transferability of gradients across different models, we initialize the perturbation of the dual example as uniform random noise. This initialization helps to explore a broader range of perturbation space and encourages the attack to discover more effective adversarial examples. Additionally, we adopt an ensemble strategy by initializing multiple dual examples, which can capture a diverse set of starting points and gradients. This ensemble approach allows us to fully explore the neighborhood of the input image and leverage the average gradient from multiple data points, leading to a more comprehensive and precise estimation of the true gradient. We provide the detailed algorithm in Appendix B.2.

**Results**. Similar to Sec. 3.3, we adopt the inverse decreasing sequences in Fig. 3 as the increasing sequence. As shown in Tab. 3, the dual example can significantly boost seven adversarial attacks. By achieving the trade-off between utilizing gradients near the benign samples and efficient updates during the later steps, the dual example can achieve much better performance than the scheduled step size. In addition, by adopting an ensemble strategy, where multiple dual examples are utilized, we further boost the adversarial transferability. The ensemble approach provides a more comprehensive

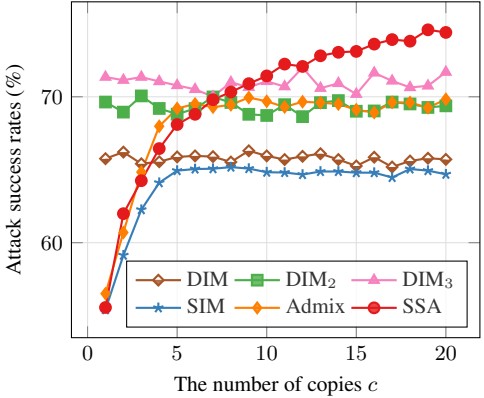

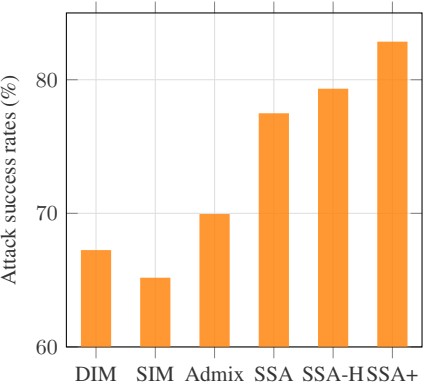

Figure 4: Attack success rates (%) when varying the number of copies.

Figure 5: Average attack success rates (%) of baselines and proposed tricks on SSA.

exploration of the neighborhood around the input image and allows for the aggregation of gradients from multiple starting points. As a result, we achieve a clear margin of improvement over the baselines, with performance gains of $26.8\%$ on I-FGSM, $12.39\%$ on MI-FGSM, $9.25\%$ on VMI-FGSM, and $10.94\%$ on GIMI-FGSM. These results validate our initial motivation and hypothesis that *focusing on the gradients of data points close to benign samples can mitigate overfitting issues and generate more transferable adversarial examples*.

### 3.5 INPUT TRANSFORMATION-BASED ATTACKS USING SPECTRAL SIMULATIONS

Input transformation-based attacks have shown superior performance by transforming the input image before gradient calculation, such as DIM (Xie et al., 2019), TIM (Dong et al., 2019), SIM (Lin et al., 2020), Admix (Wang et al., 2021a) and SSA (Long et al., 2022). Recent input transformation-based attacks often introduce a large variation on the input image for high diversity. However, this variation can introduce additional noise and uncertainty, making it challenging to obtain precise gradients for adversarial perturbation calculation. To eliminate the variance by these transformations, they adopt the average gradient of several transformed images, which can lead to unfair comparisons and biased results. Here we first investigate the impact of the number of copies on the adversarial transferability for DIM, SIM, Admix and SSA and propose a spectral-based input transformation for transferable adversarial examples.

**Hyper-parameter study**: To investigate the impact of the number of copies, we conduct the attacks using the number of copies from 1 to 20. DIM, the first input transformation-based attack, pads the resized image into $1.1\times$ size, which introduces limited diversity of transformed data. Here we further consider padding the resized image into $2\times$ and $3\times$ size, denoted as $DIM_2$ and $DIM_3$. As shown in Fig. 4, $DIM_3$ exhibits better transferability than vallina DIM. This result validates our hypothesis that using a small padding size in DIM limits the diversity of transformed data and hampers its transferability. In general, the number of copies significantly influences SIM, Admix and SSA while having little effect on DIM. This suggests that *attacks with larger variations, such as SIM, Admix, and SSA, require a larger number of copies to eliminate the variance introduced by the transformations. Furthermore, our findings indicate that a higher diversity, achieves through a larger number of copies, leads to improved transferability of adversarial examples*.

**High frequency transformation generates more transferable adversarial examples**: As shown in Fig. 4, SSA exhibits the best performance by transforming the input image in the full frequency domain. Note that a lot of work has pointed out that deep models usually focus on the low-frequency region, which is also the spectral representation of the contour and shape in the image (Long et al., 2022), while the high-frequency region captures more noise (Chang-Yanab et al., 2008). Moreover, recent research has shown that perturbing the high-frequency region can be more effective in crafting adversarial examples (Wang et al., 2020b; 2022b). Motivated by these observations, we propose SSA-H, which perturbs the high-frequency region (representing $20\%$ in the frequency domain) to augment the data. As presented in Fig. 5, SSA-H exhibits a $1.84\%$ improvement in the average attack success rate compared to vanilla SSA. This result further confirms the effectiveness of high-frequency perturbation in enhancing adversarial transferability.

**Spectral-based input transformations**: Building upon the observation that high-frequency transformations are more effective in generating transferable adversarial examples, we propose SSA+ as an extension of SSA. Specifically, SSA+ integrates conventional augmentation techniques used in input transformation-based attacks, such as scaling, dropout, and adding noise, into the spectral domain by perturbing the high-frequency domain for the attack. As shown in Fig. 5, SSA+ achieves a significant improvement of $5.36\%$ over SSA. This finding highlights the potential benefits of transforming augmentation techniques from the time domain to the frequency domain for enhancing adversarial transferability. We provide the detailed algorithm in Appendix B.3.

## 3.6 ENSEMBLE STRATEGIES

Ensemble attack has been widely recognized as an effective approach to enhance the transferability of adversarial examples, which can be generally categorized into two categories: lateral strategy and longitude strategy. Lateral strategy parallel attacks multiple models by fusing the predictions, logits, or losses of multiple models, denoted as prediction, logit, and loss based ensemble, respectively. Longitude strategy (Li et al., 2020) attacks multiple models in sequence. As revealed by the saliency map (Dong et al., 2019), different models exhibit identity and different regions of interest, which attribute to adversarial transferability across models. We provide a deep dive into different ensemble strategies using the saliency map and gradient analysis in Appendix D. To better utilize the diversity of model interests, we propose three tricks to boost the ensemble adversarial attacks.

**Gradient Alignment (GA)**: When using an ensemble strategy for crafting adversarial perturbations, one challenge arises from the variation in the regions of interest across different models. This variation leads to differences in the backpropagated gradients from the surrogate ensemble models, which can conflict with each other and hinder the effectiveness of the ensemble attack. To overcome this challenge, we propose Gradient Alignment (GA) to generate adversarial perturbations that generalize well across different models. Specifically, for each model, we calculate the cosine similarity between the gradient from this model and the mean of gradients from others. If the cosine similarity is less than 0, it indicates a conflict between the gradients, which can decrease the attack efficiency. To mitigate this conflict, we apply the Gram-Schmidt orthogonalization to the conflicted gradient before averaging the gradient for attack. We provide the detailed algorithm in Appendix B.4.

**Asynchronous Input Transformation (AIT):** Existing works (Xie et al., 2019; Dong et al., 2019; Lin et al., 2020) have integrated input transformation-based methods, such as resizing and padding, translation, scale into ensemble strategies to enhance transferability. However, these approaches typically apply the same transformations to inputs across multiple models, which may limit the input diversity and potentially hinder the overall transferability. Motivated by this, we propose a novel approach that incorporates asynchronous input transformations (such as vertical and horizontal shift, random rotations, scaling, random resizing, adding noise, and dropout) into ensemble attacks. Instead of applying aligned transformations to inputs across all models, we apply asynchronous input transformation for different individual models. By doing so, we aim to increase the input diversity within the ensemble, leading to improved transferability of the generated adversarial examples.

**Model Shuffle (MS):** In the longitude strategy, the attacker attacks multiple models sequentially, which might overfit the specific characteristics of each model and limit the transferability. To alleviate such an issue, we propose Model Shuffle (MS), which iteratively randomly shuffles the order of models during the attack process. By introducing randomness in the model order, we aim to reduce the potential bias towards specific models and promote more transferable adversarial perturbations.

**Results**. We select four models as the surrogate models: ResNet-18, ResNet-101, DenseNet-121, and MobileNet. These models are used to craft adversarial perturbations using the MI-FGSM attack with cross-entropy loss function. We set the number of attack iterations for the ensemble models to 10. Another four black-box models, *i.e.*, VGG-19, ResNeXt-50, ViT, and Swin, serve as the target models for evaluating the transferability of the adversarial perturbations generated by the surrogate models. As shown in Tab. 4, the longitude ensemble strategy outperforms the runner-up strategy with a performance improvement of $2.9\%$ in terms of adversarial transferability, which is consistent with our analysis. Among the lateral ensemble methods, GA improves the performance with an average improvement of $1.2\%$, while the AIT achieves an average improvement of $2.1\%$. MS brings a $1.6\%$ improvement in adversarial transferability for the longitude ensemble method. These results demonstrate the effectiveness and superiority of the proposed ensemble strategies in enhancing the transferability of adversarial examples.

Table 4: Attack success rates (%) using various ensemble strategies.

| Strategy | Trick | ASR |
|---|---|---|
| Loss | - | 72.4 |
| | GA | 74.3 |
| | AIT | 73.1 |
| Logit | - | 82.3 |
| | GA | 83.6 |
| | AIT | 86.5 |
| Prediction | - | 73.5 |
| | GA | 74.8 |
| | AIT | 75.0 |
| Longitude | - | 85.2 |
| | MS | 87.8 |

Table 5: Attack success rates (%) of our combined tricks and various attacks on the defense methods and Google's vision API.

| Method | Tricks | AT | FAT | HFD | RS | NRP | Google |
|---|---|---|---|---|---|---|---|
| VMI | ✗ | 74.9 | 36.1 | 84.9 | 28.2 | 70.4 | 73.0 |
| | ✓ | **98.6** | **60.8** | **99.8** | **55.3** | **88.2** | **100.0** |
| RAP | ✗ | 59.2 | 33.5 | 75.3 | 20.6 | 58.1 | 52.0 |
| | ✓ | **65.3** | **36.9** | **79.2** | **24.8** | **65.7** | **62.0** |
| NAA | ✗ | 69.5 | 34.3 | 80.2 | 25.5 | 64.6 | 59.0 |
| | ✓ | **80.2** | **61.3** | **85.7** | **35.9** | **73.1** | **71.0** |
| MIG | ✗ | 72.1 | 38.6 | 69.5 | 24.6 | 65.3 | 66.0 |
| | ✓ | **91.3** | **52.1** | **88.5** | **42.4** | **81.6** | **79.0** |
| TGR | ✗ | 75.1 | 35.3 | 82.4 | 29.5 | 73.0 | 70.0 |
| | ✓ | **97.3** | **61.5** | **95.4** | **57.6** | **91.3** | **100.0** |
| MI-CWA | ✗ | 82.7 | 45.2 | 89.1 | 39.2 | 83.5 | 83.0 |
| | ✓ | **99.1** | **62.4** | **100.0** | **59.1** | **92.8** | **100.0** |

## 4 EVALUATION ON THE COMBINED TRICKS

To further validate the effectiveness of the proposed tricks, we combine them to attack defense models and a real-world application, Google's vision API. Specifically, we select various advanced adversarial attack methods as the base strategy, including the VMI (Wang & He, 2021), RAP (Qin et al., 2022), NAA (Zhang et al., 2022), MIG (Ma et al., 2023), TGR (Zhang et al., 2023), and MI-CWA (Chen et al., 2023). Under the ensemble setting of the pool of eight surrogate models, we use these methods combined with our tricks to generate the adversarial examples and fool advanced defense methods, including adversarial training (AT) (Madry et al., 2018), high-level representation guided denoiser (HGD) (Liao et al., 2018), random smoothing (RS) (Cohen et al., 2019), and neural representation purification (NRP) (Naseer et al., 2020). For comparison, we use different methods to attack defense models without any tricks. We use different methods to generate $1,000$ adversarial examples to fool defense models and $100$ adversarial examples to deceive Google's vision API.

Tab. 5 presents the effectiveness of our combined methods compared to various baselines when attacking several advanced defense methods and Google's vision API. The use of our proposed tricks significantly enhances the attack performance of existing works against the defense models. Notably, when integrating our proposed tricks to the SOTA method MI-CWA, it has a remarkable improvement of $19.9\%$ when attacking the most powerful defense method RS. This significant improvement highlights the importance of our proposed tricks in enhancing adversarial transferability against defense methods and real-world applications. The results demonstrate the effectiveness of our proposed tricks in generating more transferable adversarial examples, allowing them to successfully bypass the advanced defense mechanisms employed by the targeted models and apply them to real-world scenes. Besides, we present the whole picture of the combination of our tricks in Fig. C4 of Appendix C.4, provide experiments of ablation study on each of the integrated tricks (see Appendix C), and integrating our tricks into more advanced ensemble frameworks (Xiong et al., 2022; Hao et al., 2022) (see Appendix D.3).

## 5 CONCLUSION

In this work, we propose a comprehensive set of tricks to enhance the adversarial transferability of existing methods. These tricks are designed for different types of attacks, including gradient-based attacks (*e.g.*, global momentum initialization, scheduled step size, and dual example), input transformation-based attacks (*e.g.*, spectral-based input transformation), and ensemble attacks (*e.g.*, gradient alignment, asynchronous input transformation, and model shuffle). Extensive experimental results validate the effectiveness of our proposed tricks, which can be combined to further improve the adversarial transferability. The proposed bag of tricks provides valuable insights and techniques to enhance the transferability of adversarial attacks and highlights the importance of understanding and exploring the factors that influence the transferability. Also, there is still remaining work to be explored, such as the optimal scheduled step size, transformations in the spectral domain, *etc*.

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

## A   EXPERIMENT SETTINGS

In the introduction, we adjust the hyper-parameters to boost the attack performance against Google's Vision API. Here, we present the details of Adjustment. For I-FGSM, we set the number of steps as 2. For MI-FGSM, we set the number of iterations as 5 with the momentum decay factor as 1.2. For PI-FGSM, we set the number of steps as 18, the step size scale factor as 0.8, and the momentum decay factor as 0.95. For VMI-FGSM, we set the number of steps as 32, the momentum decay factor as 1.0, and the number of copies as 15. For GIMI-FGSM, we set the number of iterations as 4, the number of pre-computed iterations as 9, and the momentum decay factor as 1.3.

Table A1: Parameter settings used in our research. We denote $\gamma$ as the momentum decay factor, $n_1$ as the number of random initialization, $p$ as the scaled factor for pvalue scheduled step size, $n_2$ as the number of copies for input transformation-based method, $\tau$ as the criteria to judge whether the two gradients have conflicts for gradient alignment in ensemble strategy.

| Method | Parameters |
|---|---|
| Random global initialization | $\gamma = 1.0, n_1 = 5$ |
| Scheduled step size | $p = 0.6$ |
| Dual example | $n_2 = 3$ |
| High-frequency augmentation | $c = 20, \rho = 0.2$ |
| Ensemble Strategy | $\tau = 0.1$ |

In our paper, we evaluate our tricks on 10,000 images, of which 10 images are selected from each class in ImageNet-1K. For each of the models, we use the pre-trained models provided by Pytorch trained on the ImageNet-1K dataset. We summarize the key parameters for different tricks in Tab. A1.

## B   IMPLEMENTATION DETAILS FOR OUR TRICKS

In this section, we provide additional implementation details of our tricks.

### B.1   RANDOM GLOBAL MOMENTUM INITIALIZATION

Alg. 1 outlines the implementation details of random global momentum initialization. Given a benign sample $x$ with its corresponding ground-truth label $y$, we initialize $N$ random perturbations. Each perturbation is added to a separate copy of the benign sample, resulting in $N$ parallel perturbed copies. We then apply the MI-FGSM attack to each perturbed copy for a pre-defined number of iterations $T'$. During this process, we calculate the global momentum achieved in each MI-FGSM run and compute the average global momentum as the enhanced global momentum. Afterward, we reset the perturbation to zero, set the momentum as the enhanced global momentum, and proceed with the adversarial attack using the enhanced global momentum in the subsequent iterations. The additional iterations for global momentum computation definitely increase the computation overhead and training time (around 50%). However, it is noteworthy that this increase in computation facilitates a substantial enhancement in adversarial transferability. Besides, the computation of $N$ copies is independent, which is amenable to parallel computation.

---

**Algorithm 1** Random global momentum initialization

---

**Input:** The neural network $f(\cdot)$, benign sample $x$ with the ground truth $y$, loss function $\mathcal{L}$, number of iterations $T'$ and number of random restarts $N$ for momentum initialization, number of iterations $T$ for attack, momentum decay factor $\gamma$, step size $\alpha$.
**Output:** The adversarial perturbation $\delta$.
1: **for** $n = 1$ to $N$ **do**
2:      Initialize the momentum $m_{n,0} = 0$, and randomly initialize $\delta_{n,0}$
3:      **for** $t = 1$ to $T'$ **do**
4:          $m_{n,t} \leftarrow \nabla_x \mathcal{L}(f(x + \delta_{n,t-1}), y) + \gamma \cdot m_{n,t-1}$
5:          $\delta_{n,t} \leftarrow \delta_{n,t-1} + \alpha \cdot \text{sign}(m_{n,t})$
6:      **end for**
7: **end for**
8: Initialize $\delta_0$ with 0, and the momentum $m_0$ with $\frac{1}{N}\sum_{n=1}^N m_{n,T'}$
9: **for** $t = 1$ to $T$ **do**
10:      $m_t \leftarrow \nabla_x \mathcal{L}(f(x + \delta_{t-1}), y) + \gamma \cdot m_{t-1}$
11:      $\delta_t \leftarrow \delta_{t-1} + \alpha \cdot \text{sign}(m_t)$
12: **end for**
13: **return** $\delta_T$

---

## B.2 DUAL EXAMPLE

We present the implementation of dual examples with ensemble strategy in Alg. 2. We initialize $\delta_0 = 0$ and $N$ dual adversarial perturbations with random initialization, $\{\delta_{n,0}^{dual}\}_{n=1}^N$. For each dual example $\delta_{n,t}^{dual}$ at the $t$-th iteration, we compute the gradient $g_{n,t}$ and adopt an increasing scheduled step size $\alpha_n$ to update the dual example through I-FGSM optimization. Then, we calculate the average gradient $\frac{1}{N}\sum_{n=1}^N g_{n,t}$ and update the adversarial perturbation $\delta_t$ using MI-FGSM optimization. The algorithm iterates for a specified number of iterations $T$ to refine the adversarial perturbation. Finally, the adversarial example is obtained by adding the refined perturbation $\delta_T$ to the original input $x$. The utilization of dual examples with an ensemble strategy allows for a more comprehensive exploration of the perturbation space and enhances the effectiveness of adversarial attacks. When $N = 1$, **it degenerates to the vanilla dual example method**. Besides, it should be noted that the computation of each dual example in the ensemble strategy is independent and parallel, which is amenable for GPU to efficient computation.

---

**Algorithm 2** Dual example with ensemble strategy

---

**Input:** The neural network $f(\cdot)$, benign sample $x$ with the ground truth $y$, loss function $\mathcal{L}$, number of iterations $T$, number of dual examples $N$, momentum decay factor $\gamma$, increasing scheduled step size sequence $\{\alpha_t\}_{t=1}^T$.
**Output:** The adversarial perturbation.
1: Initialize $\{\delta_{n,0}^{dual}\}_{n=1}^N$ using the random initialization, and $\delta_0 = 0$
2: Initialize the momentum $m_0$ with 0
3: **for** t=1 to T **do**
4:      **for** n=1 to N **do**
5:          $g_{n,t} \leftarrow \nabla_x \mathcal{L}(f(x + \delta_{n,t-1}^{dual}), y)$
6:          $\delta_{n,t}^{dual} \leftarrow \delta_{n,t-1}^{dual} + \alpha_t \cdot \text{sign}(g_{n,t})$
7:      **end for**
8:      $m_t \leftarrow \frac{1}{N}\sum_{n=1}^N g_{n,t} + \gamma \cdot m_{t-1}$
9:      $\delta_t \leftarrow \delta_{t-1} + \alpha_t \cdot \text{sign}(m_t)$
10: **end for**
11: **return** $\delta_T$

---

## B.3 SPECTRAL-BASED INPUT TRANSFORMATIONS

We denote three spectral-based transformations: scale, add noise and dropout. The implementations of these transformations are described as follows:

**Scale**: We perform a scaling operation on the spectral representation of images by multiplying its frequency values with a random coefficient $\alpha \in (0, 1)$.

**Add Noise**: We introduce uniform random noise to perturb the frequency values of the spectral representation of images.

**Dropout**: We randomly set $10\%$ frequency values in the spectral representation of images to 0.

Notably, these transformations exclusively affect the top $20\%$ of the high-frequency region while leaving the remaining frequency regions unaffected. By focusing on the high-frequency part of the image, these transformations introduce controlled perturbations that can enhance the transferability of adversarial examples.

### B.4 GRADIENT ALIGNMENT

---
**Algorithm 3** Ensemble adversarial attack with gradient alignment

---
**Input:** $N$ surrogate neural networks $f_n(\cdot)$, $n = 1, 2, ., , , N$, benign sample $x$ with the ground truth $y$, loss function $\mathcal{L}$, number of iterations $T$, and step size $\alpha$.
**Output:** The adversarial perturbation $\delta$
1: Initialize the adversarial perturbation $\delta_0 = 0$
2: **for** $t = 1$ to $T$ **do**
3:      **for** $n = 1$ to $N$ **do**
4:          $g_n \leftarrow \nabla_x \mathcal{L}(f_n(x + \delta_{t-1}), y)$
5:          $g_{\text{avg}} = \frac{1}{N-1}\left[\sum_{j=1}^{N} \nabla_x \mathcal{L}(f_j(x + \delta_{t-1}), y) - g_n\right]$
6:          **if** $\text{sign}(g_n) \cdot \text{sign}(g_{avg}) < 0$ **then**
7:             $g_n \leftarrow g_n - \frac{g_n \cdot g_{\text{avg}}}{\|g_{\text{avg}}\|_2^2} g_{\text{avg}}$
8:          **end if**
9:      **end for**
10:      $\delta_t \leftarrow \delta_{t-1} + \alpha \cdot \text{sign}(\sum_{n=1}^{N} g_n)$
11: **end for**
12: **return** $\delta_T$

---

We present the implementation of the gradient alignment strategy in Alg. 3. Given the gradients of the adversarial perturbation obtained from multiple surrogate models, the algorithm iterates through each gradient. For each gradient, it calculates the inner product between the sign value of that gradient and the mean of the remaining gradients. This inner product value serves as an indicator of whether a conflict arises between the gradients. If the inner product value is less than 0, indicating a conflict, the algorithm orthogonalizes the conflicting gradient. By iterating through all gradients and aligning them to remove conflicts, the efficiency and effectiveness of the ensemble attack are enhanced.

## C ABLATION STUDY ON TRICKS

### C.1 MOMENTUM INITIALIZATION

In our momentum initialization strategy, we employ multiple copies with random initialization to assist the adversarial perturbation in escaping local maxima during the momentum initialization. This enables the discovery of a better global momentum. We specifically select 5 copies for this purpose. Here, we also explore the impact of varying the number of copies on the performance of adversarial transferability.

The ablation study on the number of copies for our momentum initialization strategy is presented in Fig. C1. The results demonstrate that increasing the number of copies leads to an improvement in the success rate of adversarial attacks across all studied attack methods. When comparing our proposed strategy with only 1 copy, which corresponds to the GIMI-FGSM approach, a clear performance gap of $9.82\%$ is observed for various attacks when using 10 copies. This finding further supports our argument that achieving optimal global momentum requires thoroughly exploring the entire neighborhood.

## C.2 DUAL EXAMPLE

In our study, we incorporate the ensemble strategy (5 copies) to enhance the performance of dual examples. Here, we conduct an ablation study to assess the impact of the number of copies on adversarial transferability. Fig. C2 presents the results of the ablation study on the number of copies. It is evident that as the number of copies increases, the attack success rate also increases. When compared to the vanilla dual example strategy with only $N = 1$ copy, each adversarial attack demonstrates a significant improvement of $9.1\%$ on average with an increase in the number of copies to 10 ($N = 10$). These findings emphasize the effectiveness of conducting a comprehensive exploration in the vicinity of the benign sample to enhance adversarial transferability. By utilizing multiple copies of dual examples, we can capture a wider range of perturbation variations, leading to improved performance in generating transferable adversarial examples.

## C.3 SPECTRAL-BASED INPUT TRANSFORMATION

In our study, we utilize spectral-based input transformations focusing on high-frequency components for adversarial attacks. In our paper, we specifically select the top $20\%$ of the high-frequency components. Here we also conduct an additional ablation study to investigate the effect of varying the ratio of high frequency on the performance of adversarial transferability.

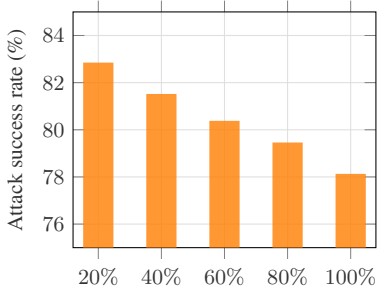

Consistent with the settings in our paper, Fig. C3 depicts the results of the ablation study. We can observe a decreasing trend in the adversarial attack success rate as the ratio of high-frequency components increases. Comparing the results of applying spectral-based input transformations on the full frequency range versus the top $20\%$ frequency range, we note a performance drop of $3.72\%$ in terms of the adversarial attack success rate.

Figure C3: Ablation study on the selection of the ratio for high frequency in spectral-based adversarial attacks.

These findings provide further evidence of the efficacy of utilizing high-frequency components to enhance adversarial transferability.

## C.4 COMBINATION OF TRICKS

In our paper, we integrate the combination of tricks into the ensemble strategies to attack the defense methods and show the effectiveness of our tricks. We use Fig. C4 to demonstrate the whole picture of the combination of tricks in adversarial attacks. Here, we conduct an ablation study on the combination of tricks by iterative removing each of the tricks.

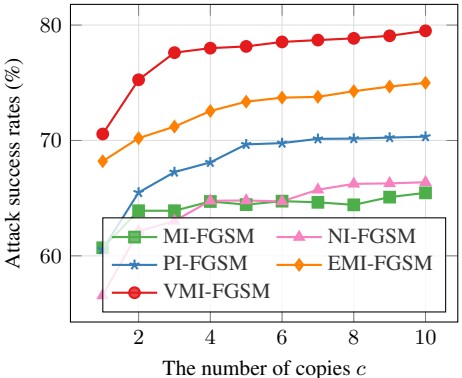
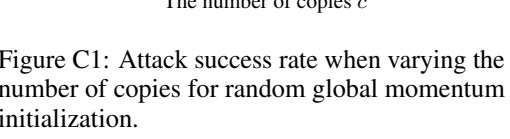

Figure C1: Attack success rate when varying the number of copies for random global momentum initialization.

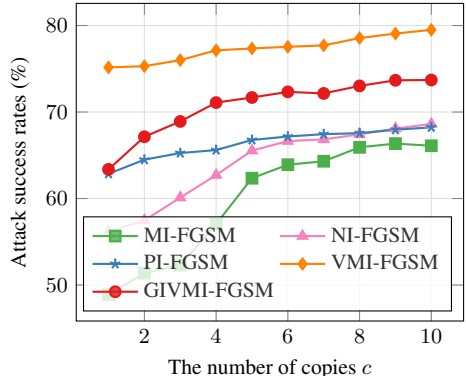

Figure C2: Attack success rate when varying the number of copies for the dual ensemble example strategy.

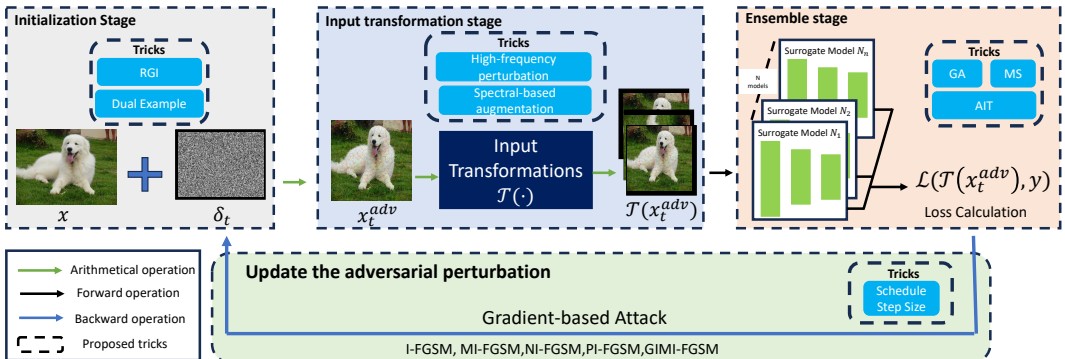

Figure C4: Overview of the combination of our proposed tricks. We use the random global initialization (RGI) and dual example strategy in the initialization stage. We use high-frequency perturbation with spectral-based augmentation to boost the performance of input transformation-based attacks. We integrate the gradient alignment (GA), asynchronous input transformation (AIT), and model shuffle (MS) into the ensemble-based attacks.

Table C2: Attack success rate of the ensemble attack using a bag of tricks when removing each component. We denote H as the high-frequency transformation, D as the dual example, S as the scheduled step size, R as the random global initialization, GA as the gradient alignment, MS as the model shuffle, and AIT as the asynchronous input transformation. We use '-' to represent the removing operation.

| Methods | AT | FAT | HGD | RS | NRP |
|---------|------|------|------|------|------|
| ours | 98.6 | 60.8 | 99.8 | 55.3 | 88.2 |
| ours-H | 98.0 | 59.5 | 99.2 | 55.1 | 87.6 |
| ours-H-D | 94.3 | 52.9 | 97.1 | 49.6 | 84.7 |
| ours-H-D-S | 93.2 | 50.2 | 96.5 | 48.0 | 83.5 |
| ours-H-D-S-R | 91.7 | 45.4 | 96.1 | 40.8 | 81.4 |
| ours-H-D-S-R-GA | 91.0 | 44.3 | 95.9 | 39.4 | 80.6 |
| ours-H-D-S-R-GA-MS | 90.9 | 43.9 | 95.8 | 38.1 | 80.2 |
| ours-H-D-S-R-GA-MS-AIT | 90.6 | 43.4 | 95.8 | 37.2 | 79.8 |

As shown in Tab. C2, we can see a clear performance drop with removing each of the integrated tricks. But the combination of remaining tricks still presents high attack success rate against different defense methods.

## C.5 ORTHOGONALITY STUDY

We conduct experiments on the orthogonality study of proposed tricks. We pair up the proposed tricks two by two and evaluate the attack performance of the combination. The studied tricks include random global momentum initialization (RGI), dual example with scheduled step size (DS), high-frequency spectral transformation (HT), gradient alignment (GA), asynchronous input transformation (AIT), and model shuffle (MS). We use the MI-FGSM as the optimization method for attack under the ensemble settings in our paper. We report the average attack success rates against the defense models for study, including the AT, FAT, HGD, RS, and NRP. The results are presented in Tab. C3.

From the results, we can see the integration of different tricks can improve the performance further. For example, while the attack success rate using RGI is 48.2% and using DS is 49.3%, the combination of RGI and DS is 51.5%. It supports our argument that the tricks are orthogonal to each other in boosting performance.

Table C3: Average attack success rates (%) of the combination of our combined tricks on the defense methods.

| Tricks | RGI | DS | HT | AIT | MS | GA |
|--------|-----|-----|------|------|------|------|
| RGI | 48.2 | 51.5 | 53.4 | 49.3 | 48.5 | 48.7 |
| DS | - | 49.3 | 55.9 | 51.8 | 50.6 | 53.9 |
| HT | - | - | 50.2 | 57.0 | 53.4 | 51.8 |
| AIT | - | - | - | 48.5 | 49.3 | 49.0 |
| MS | - | - | - | - | 47.6 | 48.5 |
| GA | - | - | - | - | - | 48.1 |

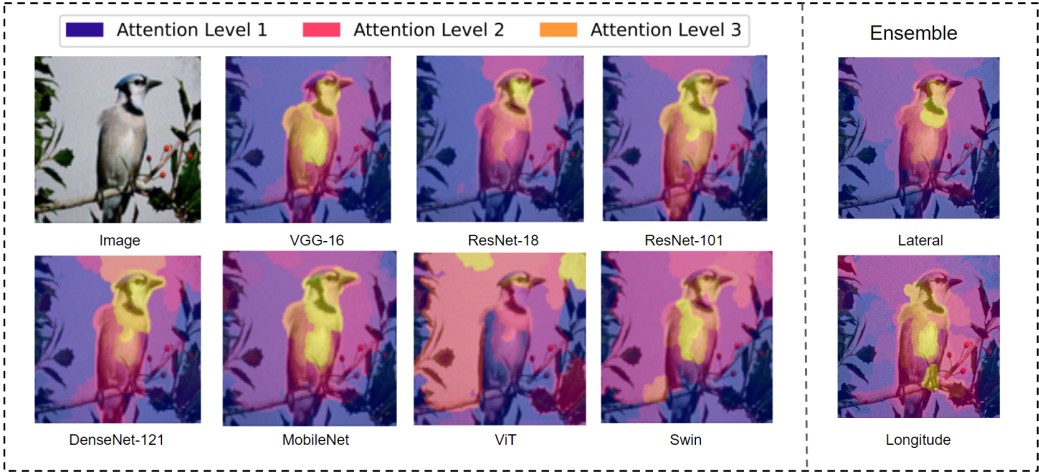

Figure D5: Visualization of the saliency maps for different models on the same input image. The strength of the colors in the maps represents the importance of the features that influence the classification decision (orange > red > blue).

# D A DEEP DIVE INTO THE ENSEMBLE STRATEGIES

## D.1 SALIENCY MAPS OF DIFFERENT MODELS AND ENSEMBLE STRATEGIES

Saliency maps represent the areas of the image that have the most influence on the models' decision-making process, with different levels of importance represented by different colors. To explore the identities and differences across various models, we present the saliency maps generated by XRAI on different models and ensemble strategies in Fig. D5. We can see that most models exhibit similar regions of interest, particularly focusing on the bird's head. This indicates the presence of adversarial transferability, where the models share a common vulnerability to adversarial perturbations in that region. However, there are notable differences among the models as well. For example, MobileNet emphasizes the bird's body, while ViT pays more attention to the background. These differences in areas of interest can help explain the variations in adversarial transferability observed across different models. We also compare the saliency maps generated using the logit-based lateral strategy and the longitude-based ensemble strategy. We can observe that the region of interest identified by the longitude ensemble strategy is larger than that of the lateral ensemble strategy. This suggests that the longitude ensemble strategy is more effective in capturing important regions for adversarial transferability, resulting in better performance.

## D.2 GRADIENT ANALYSIS OF DIFFERENT ENSEMBLE STRATEGIES

The objective function of each ensemble framework can be formulated as

- logit- or prediction-based ensemble: $\max_\delta \mathcal{L}(x + \delta, y) = -1_y \cdot \log(\sum_{k=1}^{K} w_k m_k(x + \delta))$,
- loss-based ensemble: $\max_\delta \mathcal{L}(x + \delta, y) = \sum_{k=1}^{K} (-1_y \cdot \log(m_k(x + \delta)))$,
- longitude strategy: $\max_\delta \mathcal{L}_k(x + \delta, y) = -1_y \cdot \log(m_k(x + \delta))$, $k \in \{1, \cdots, K\}$.

Here $w_k$ is the weight for $k$-th model, $m_k$ is the $k$-th model output (logits or prediction), and $1_y$ is the one-hot encoding of the ground-truth label $y$ of $x$.

We can calculate the gradient w.r.t. $z_{k,c}$ for loss-based ensemble as $p_{k,c} - y_c$, for logit-based ensemble as $\frac{1}{K} \sum_{m=1}^{K} p_{m,c} - y_c$, and for prediction-based ensemble as $\left( \frac{1}{K} \sum_{m=1}^{K} p_{m,c} - y_c \right) (1 - p_{k,c}) p_{k,c}$, where $p_{k,c} = \frac{e^{z_{k,c}}}{\sum_i e^{z_{k,i}}}$ is the probability output for class $c$ of model $k$, and $z_{k,c}$ is the logit output for class $c$ of model $k$. Compared with the loss-based ensemble, the $k$-th model's backpropagated gradient using the logit- and prediction-based ensemble strategy also infuse with other models' meaningful perception information, thus enhancing the adversarial transferability. The term $(1 - p_{k,c}) p_{k,c}$ in prediction-based ensemble attacks is less than 1, which leads to a weaker gradient than logit-based ensemble attacks. Thus, the logit-based ensemble yield a much stronger gradient than the prediction- and loss-based ensemble.

Then, compared with attacking multiple models in parallel which optimizes multiple targets together, it is more efficient to craft stronger adversarial examples by attacking one model one time, which corresponds to using the greedy strategy for multi-object optimization, with a limited number of iterations. Based on our analysis, the rank of efficiency of different ensemble attacks is

$$\text{longitude} > \text{logit-based} > \text{prediction-based} > \text{loss-based,} \tag{2}$$

which is also verified by our experiment results.

## D.3 INTEGRATED INTO OTHER ENSEMBLE METHODS

In our paper, we integrate the combination of tricks into three general ensemble strategies. There are also some novel ensemble strategies proposed to enhance the performance of ensemble attacks, such as the stochastic variance reduced ensemble (SVRE) and the stochastic serial attack (SSA) adversarial attack. SVRE aims to reduce the variance of the gradients of each model for a more transferable ensemble attack. SSA is a longitudinal-based ensemble attack that randomly selects a subset of models for attack in each iteration. In contrast, our approach utilizes all models for attack in each iteration.

To verify the effectiveness of our proposed tricks, we integrated them into the SVRE and SSA methods, respectively. The results, as shown in Table D4, demonstrate that the integration of our

Table D4: Attack success rate of different methods against defense methods. Here, we denote the combination of tricks in our paper as ours, and integrate the combination of our tricks into two ensemble strategies SVRE and SSA as SVRE+ours and SSA+ours, respectively.

| Methods | AT | FAT | HGD | RS | NRP | Swin-L | XCiT-L12 |
|---------|------|------|------|------|------|--------|----------|
| ours | 98.6 | 60.8 | 99.8 | 55.3 | 88.2 | 37.4 | 53.9 |
| SVRE | 92.3 | 45.7 | 97.2 | 38.1 | 81.6 | 20.5 | 38.7 |
| SVRE+ours | 99.9 | 62.7 | 99.7 | 58.5 | 90.4 | 38.1 | 55.6 |
| SSA | 90.8 | 42.5 | 95.3 | 37.2 | 80.6 | 14.6 | 35.2 |
| SSA+ours | 97.3 | 58.2 | 99.1 | 54.0 | 86.5 | 37.4 | 50.4 |

proposed tricks further enhances the performance of both SVRE and SSA. This indicates that the tricks are complementary and can significantly improve the transferability of ensemble attacks.

Furthermore, the average attack success rate of our proposed approach is better than SSA, which suggests that utilizing all models for attack in each iteration leads to better strategies for achieving higher transferability.

# E    ADDITIONAL EXPERIMENTS ON RESNET-18

In our paper, we conduct experiments on VGG-16 to verify the effectiveness of the proposed tricks, including the random global momentum initialization, scheduled step size, dual example, and spectral-based input transformations. Here, we further conduct experiments on ResNet-18 to verify the generality of the proposed tricks among different models.

## E.1    RANDOM GLOBAL MOMENTUM INITIALIZATION

Table E5: Average attack success rate of momentum-based attacks with GI or RGI for global momentum initialization, including MI-FGSM, NI-FGSM, PI-FGSM, EMI-FGSM, and VMI-FGSM. We report the performance under the default setting ($T = 10$).

|  | MI-FGSM | NI-FGSM | PI-FGSM | EMI-FGSM | VMI-FGSM |
|------|---------|---------|---------|----------|----------|
| Ori. | 62.1 | 63.7 | 65.6 | 69.2 | 76.5 |
| GI | 67.6 | 63.6 | 66.6 | 75.1 | 77.5 |
| RGI | **70.5** | **70.6** | **72.9** | **77.6** | **83.4** |

Following the same setting in our paper, Tab. E5 presents the results of random global momentum initialization. It can be observed that global initialization has a minor or negative effect on the adversarial transferability of a few baselines, including NI-FGSM, PI-FGSM, and VMI-FGSM. In contrast, the RGI method significantly improves the adversarial transferability for all the baselines, surpassing the GI method with a mean attack success rate of $4.92\%$. These results provide further confidence in supporting our argument that proper initialization of the global momentum requires a comprehensive exploration of the neighborhood. The effectiveness of the RGI method in enhancing the adversarial transferability across various baselines demonstrates the importance of initializing the momentum in a way that encourages the discovery of more effective perturbations.

## E.2    SCHEDULED STEP SIZE

The results presented in Table E6 are consistent with the conclusions of our paper, demonstrating that the scheduled step size can improve the adversarial transferability of all five attacks. However, it is also evident that the choice of the scheduled sequence can have a different impact on the adversarial transferability depending on the model being targeted. For instance, the exp sequence has a negative effect on the performance when targeting VGG-16, resulting in a performance drop of $8.1\%$. However, it has a positive impact on the performance when targeting ResNet-50, with an improvement of $0.6\%$. This observation highlights the need for further study on the selection of scheduled step sizes for different models and different attack methods.

Table E6: Average attack success rate of gradient-based attacks using various scheduled step sizes, including I-FGSM, MI-FGSM, PI-FGSM, VMI-FGSM, and GIMI-FGSM. We report the performance under the default setting ($T = 10$).

|        | I-FGSM | MI-FGSM | PI-FGSM | VMI-FGSM | GIMI-FGSM |
|--------|--------|---------|---------|----------|-----------|
| Ori.   | 41.4   | 62.1    | 65.6    | 76.5     | 67.5      |
| log    | 42.9   | 62.7    | **66.8**| **78.9** | **69.2**  |
| linear | **43.8**| 63.4   | 66.4    | 78.1     | 68.3      |
| exp    | 42.9   | **62.9**| 66.2    | 75.1     | 66.2      |
| pvalue | 40.5   | 62.2    | 64.1    | 75.6     | 67.2      |

### E.3 DUAL EXAMPLE

Table E7: Attack success rate when applying dual example w/wo ensemble strategy using various increasing sequences as step size to various attacks, *i.e.*, I-FGSM, MI-FGSM, PI-FGSM, VMI-FGSM, and GIMI-FGSM.

|        | I-FGSM     | MI-FGSM    | PI-FGSM    | VMI-FGSM   | GIMI-FGSM |
|--------|------------|------------|------------|------------|-----------|
| N/A    | 41.4       | 62.1       | 65.6       | 76.5       | 67.6      |
| Ori.   | 52.8/67.3  | 64.5/69.2  | 67.6/70.2  | 79.6/80.5  | 70.9/74.1 |
| log    | 54.0/66.5  | 65.2/69.6  | 63.9/68.9  | 81.3/81.9  | 71.4/74.3 |
| linear | 53.7/66.7  | 64.7/69.1  | 66.3/69.5  | 82.0/83.0  | 72.3/74.5 |
| exp    | 53.1/65.9  | 66.1/69.3  | 64.3/69.2  | 79.8/79.9  | 70.9/73.3 |
| pvalue | 54.1/66.7  | 62.5/68.9  | 68.9/70.0  | 77.7/79.4  | 72.1/74.1 |

Following the same experimental settings in our main text, we have evaluated the performance of the dual example method on ResNet-18. The results, presented in Tab. E7, demonstrate clear improvements over the baseline methods. Compared to the baselines, our dual example approach achieves significant performance gains on ResNet-18. Specifically, we observe improvement margins of $25.9\%$ on I-FGSM, $7.5\%$ on MI-FGSM, $4.3\%$ on VMI-FGSM, and $6.9\%$ on GIMI-FGSM. These results further demonstrate the effectiveness of the dual example and highlight the necessity to fully utilize the gradient near the benign samples to boost the adversarial transferability.

### E.4 SPECTRAL-BASED INPUT TRANSFORMATION

Following the same experimental settings as described in our main text, we have conducted experiments on ResNet-18 to evaluate the spectral-based input transformation methods. The results, depicted in Fig. E6, differ from those obtained with VGG-16. Notably, the performance of SSA is worse than that of Admix in terms of adversarial transferability. This highlights the significance of adopting various models for a thorough evaluation. By contrast, when augmenting only the top $0.2\%$ of the high-frequency components, SSA-H demonstrates an improvement of $0.58\%$ compared to SSA. Our proposed method, SSA+, surpasses the runner-up method Admix by a significant margin of $5.15\%$, achieving the best performance in terms of adversarial transferability on ResNet-18. This indicates the superiority and robustness of SSA+ in enhancing adversarial transferability.

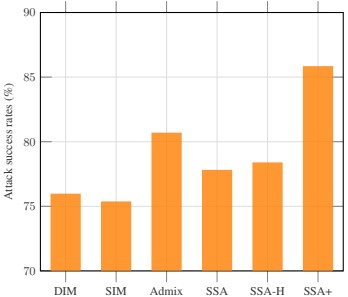

Figure E6: Comparison between baselines and proposed trick, SSA+, on the best adversarial attack success rate.

## F ADDITIONAL EXPERIMENTS ON DENSENET-121 AND VIT

Here, we also provide numeric results on DenseNet-121 and ViT to give a comprehensive study on the performance of proposed tricks.

We present results of the random global initialization in Tab. F9, the dual example in Tab. F9, the high-frequency augmentation in Tab. F10.

Table F8: Average attack success rate of momentum-based attacks with GI or RGI for global momentum initialization, including MI-FGSM, NI-FGSM, PI-FGSM, EMI-FGSM, and VMI-FGSM. We report the performance when taking the DenseNet121/ViT as the surrogate models, respectively.

| Method | MI-FGSM | NI-FGSM | PI-FGSM | EMI-FGSM | VMI-FGSM |
|--------|---------|---------|---------|----------|----------|
| Ori. | 79.7/66.2 | 82.3/66.9 | 82.7/65.8 | 88.2/72.4 | 88.5/73.1 |
| GI | 84.9/72.2 | 84.8/70.2 | 85.8/68.6 | 92.6/80.5 | 85.5/76.0 |
| RGI | **88.8/75.3** | **89.8/74.4** | **90.8/75.7** | **94.3/83.4** | **91.7/80.4** |

Table F9: Average attack success rate of the dual example strategy using various increasing sequences as step size to various attacks. The surrogate models are set as DenseNet121/ViT, respectively. We denote "N/A" as vanilla adversarial attacks without any tricks.

| Method | I-FGSM | MI-FGSM | PI-FGSM | VMI-FGSM | GIMI-FGSM |
|--------|--------|---------|---------|----------|-----------|
| N/A | 61.4/45.6 | 79.6/66.2 | 82.7/68.8 | 88.4/73.1 | 83.2/70.6 |
| Ori. | 71.3/63.1 | 82.9/68.1 | 84.4/70.4 | 92.7/78.3 | 86.7/73.4 |
| log | 72.1/**65.7** | 85.1/69.5 | 84.2/67.9 | 94.2/78.7 | **87.4**/74.1 |
| linear | 70.2/64.2 | **86.7**/71.6 | 83.5/71.3 | **95.7**/79.1 | 86.9/73.9 |
| exp | 68.9/60.9 | 83.4/70.0 | 80.4/65.1 | 92.1/75.6 | 85.3/73.0 |
| pvalue | **72.4**/64.4 | 85.5/**72.3** | **86.8/71.7** | 95.2/**80.5** | 86.4/**75.7** |

Table F10: Average attack success rate of different input transformation-based methods.

| Surrogate Models | TIM | DIM | SIM | Admix | SSA | SSA+ |
|------------------|-----|-----|-----|-------|-----|------|
| DenseNet121 | 77.6 | 90.1 | 90.1 | 91.5 | 93.0 | 97.8 |
| ViT | 57.1 | 75.2 | 69.9 | 71.3 | 77.4 | 81.8 |

The results of the random global initialization are shown in Tab. F9. The proposed tricks enhance the adversarial transferability with an average of $2.44\%$ versus the runner-up method.

The results of the dual example using different scheduled step sizes are shown in Tab. F9. It can be seen that the application of a dual example brings an average attack success rate improvement of $4.5\%$ and $5.8\%$ when taking DenseNet121 and ViT as the surrogate models, respectively. Besides, a proper selection of the scheduled step size can further boost the attack success rates up to $2.5\%$ and $4.2\%$ for DenseNet121 and ViT, respectively.

The results of the high-frequency augmentation are shown in Tab. F10. The proposed SSA+ surpasses the runner-up method with $4.8\%$ on DenseNet121 and $4.4\%$ on ViT.

## G  DISCUSSION ON THE HYPER-PARAMETER STUDY

The hyper-parameter study of Section 3 in our paper is an empirical study suggesting that adversarial transferability is much more sensitive to some basic settings than we thought. We hope to use experiments on these facts to appeal to more concerns on the overlooked confounders when benchmarking attacks. They provide practical guidance for real-world applications and motivate the design of our proposed tricks.

Although empirical research has shed light on adversarial transferability, theoretical analysis (Demontis et al., 2019; Zhu et al., 2022), especially with respect to the impact of hyperparameters, is less developed but crucial (Bose et al., 2020). Adversarial examples are typically designed using a surrogate model and then applied to attack the target model (Yang et al., 2021; Zhao et al., 2023; Zhang et al., 2024; Wang & Farnia, 2023), with a general theoretical focus on the surrogate model, the optimization scheme (Yang et al., 2021), and the target model (Zhao et al., 2023). However,

Table H11: Attack success rates (%) of integrating our tricks into MI-FGSM (MI) on the audio-visual models. We use the audio-visual model with ResNet-18 as the backbone to generate adversarial examples and attack the other two models.

| Audio-Visual Models | ResNet-18 | VGG-16 | ResNet-34 |
|---|---|---|---|
| MI | 100.0 | 39.5 | 40.6 |
| MI+RGI | 100.0 | 43.7 | 42.8 |
| MI+Scheduled step size | 100.0 | 42.1 | 41.9 |
| MI+Dual example | 100.0 | 45.2 | 46.9 |

theoretical work that specifically addresses the optimization scheme in adversarial transferability is limited.

Our study concentrates on the optimization aspects of this process, such as initialization, step size, and iteration count. Building on the perspective of adversarial example generation as an optimization process, coupled with recent theoretical advances in optimization, we believe that our findings can stimulate further theoretical exploration in this area. For instance, our observation that increasing step size improves transferability raises intriguing theoretical questions:

1. Loss Landscape Flatness: In the context of discussions on the sharpness of the loss landscape and flat optima (Li et al., 2018; Foret et al., 2020; Izmailov et al., 2018), particularly the theoretical insight into sharpness and stability analysis, indicating that a large learning rate is beneficial for a wider minima and smaller sharpness (Wu et al., 2018). A question naturally arises: Could a larger step size lead to a flat optimum with a tighter lower bound or looser upper bound of adversarial transferability? This inquiry could inspire novel strategies inspired from loss landscape flatness, potentially leading to improved bound for transferability comparing to previous work (Yang et al., 2021).

2. PAC-Bayes framework: Given the theoretical discussions on increasing learning rate schedule (Li & Arora, 2019), especially a PAC-Bayes generalization bound related to learning rate (He et al., 2019), there is another question: How does step size impact the bound of adversarial transferability? This could lead to new adversarial transferability bounds related to step size inspired by the PAC-Bayes framework or novel algorithms with a theoretically motivated step size schedule.

We hope our empirical findings will inspire deeper theoretical investigations into adversarial transferability, bridging the gap between practical experimentation and theoretical insights.

## H ATTACKING REAL-WORLD AUDIO-VISUAL MODELS

Following previous work (Tian & Xu, 2021), we conduct an experiment on attacking audio-visual models in the event classification task. For the surrogate model, we use the model that uses ResNet-18 for both audio and visual encoders and the sum operation as the fusion module for features of audio and visual modalities. The targeted model involves two models, which, respectively use VGG-16 and ResNet-34 as the visual and audio encoders. We use the audio-visual model with the ResNet-18 as the backbone to craft 200 adversarial examples and attack the other two models. We use MI-FGSM to craft the adversarial examples and respectively apply the proposed random global initialization (RGI), scheduled step size, and dual example strategies to MI-FGSM. The results are reported in H11.

From the results, we can see a clear improvement (3.8% on average) by integrating the tricks into MI-FGSM. It shows the scalability of our tricks in real-world audio-visual applications.

