# OpenReview forum: "Bag of Tricks to Boost Adversarial Transferability"
_ICLR.cc/2024/Conference — Submitted to ICLR 2024_

### Official Review · Reviewer_A7nY · 2023-10-29

**Soundness:** 2 fair
**Presentation:** 3 good
**Contribution:** 2 fair
**Rating:** 6
**Confidence:** 4

**Summary:**

This work investigates the potential tricks to improve the adversarial transferability of existing works. Specifically, the authors study number of iterations, scale factor, momentum decay factor, momentum initialization, step size scheduling, dual example, input transformation, and ensemble. The evaluation covers various baselines with their variants on ImageNet.

**Strengths:**

1. The paper is well-organized and easy to follow.
2. The evaluation is extensive, including both CNNs and ViTs, and covers several important aspects that could influence adversarial transferability.
3. The results could benefit the adversarial transferability community.

**Weaknesses:**

1. Lack of in-depth analysis. Although the authors provide extensive evaluation of various techniques, it is difficult to see the in-depth analysis to provide more insights. For example, the authors discuss the hyper-parameters study in Section 3.1. Given different patterns, how these patterns inspire new research and why these patterns exist deserve more discussion instead of simple optimal hyper-parameters takeaways.
2. Since the authors discuss various aspects that could influence adversarial transferability, it is important to discuss the orthogonality and the effectiveness of different combinations. Although part of the results is shown in Table C2, there are no comprehensive results as well as valuable analysis.
3. The covered baselines are mainly before 2022. However, there exist many recent works on adversarial transferability, such as [a, b, c, d].

[a]. Transferable Adversarial Attack for Both Vision Transformers and Convolutional Networks via Momentum Integrated Gradients. ICCV 2023.

[b]. Transferable Adversarial Attacks on Vision Transformers with Token Gradient Regularization. CVPR 2023.

[c]. Improving Adversarial Transferability via Neuron Attribution-based Attacks. CVPR 2022.

[d]. Boosting the transferability of adversarial attacks with reverse adversarial perturbation. NeurIPS 2022.

**Questions:**

1. Please provide more in-depth analysis to clarify the contribution.
2. Please provide more evaluation and analysis of orthogonality and the effectiveness of different combinations.
3. Please involve more comparison and discussion with recent work.

---

> ### Author Response · Authors · 2023-11-20
>
> ### Responses to Weakness&Questions   1
> > **A1**: [**Analysis on the hyper-parameter study**] As  you suggested, we have revised our paper to present a more in-depth analysis of our hyper-parameter study, which can be summarized as follows,
> > *  [Number of iterations T] (1) The results of I-FGSM in Fig. 2a by varying the number of iterations suggest an up-to-down trend in attack success rates, **indicating the importance of gradients near the input image in crafting transferable adversarial examples.** (2) GIMI-FGSM performs best at around T = 5, leveraging global momentum computed using the first several iterations to bypass nearby local optima effectively. However, its attack success rate diminishes with larger T values, indicating that the effectiveness of this momentum diminishes against distant local optima.
> >
> > *  [The scale factor for step size] We observed a notable performance boost with an optimal step size range (Fig. 2b). **An adequately large step size can be utilized to skip over the local optima and boost the adversarial transferability**, which can be verified by I-, MI-, NI, etc. The excessively large step size can overlook the optima and cause inefficient optimization under $l_∞$ constraint, thus harming attack performance, which can be observed in PI- and EMI-FGSM.
> >
> >
> > *  [The decay factor for momentum]  By varying the momentum decay factor (Fig. 2c), we can see a significant performance change in different attacks. The decay factor balances the historical momentum and novel gradient, **adjusting which to an appropriate ratio can stabilize the update directions**, thus better searching for the direction towards optima to boost the adversarial transferability.
> >
> > [**Takeaways**] The results of the hyper-parameter study suggest that adversarial transferability is much more sensitive to some basic settings than we thought. We appeal to more concerns on the overlooked confounders when benchmarking attacks.
> >
> >[**Inspiration**] We are motivated by the insights deriving from the study on the number of iterations and scale factor for step size to design the scheduled step size and dual example strategy, thus better utilizing the near-sample gradients for better adversarial transferability. We are motivated by the analysis of GI-MIFGSM and the study of the momentum decay factor to design the random global momentum initialization technique to find a better initialization value for momentum, thus stabilizing the optimization direction towards the optima. **However, our proposed tricks are just simple utilization of these insights, which remain more in-depth theoretical analysis and practice exploration for future work.**

---

> ### Author Response · Authors · 2023-11-20
>
> ### Responses to Weakness&Questions   2-3
> > **A2**:  As you suggested, we provide analysis and evaluation as follows,
> > [**Analysis**]  In our appendix (Fig. C4), we categorize the adversarial attack process into four distinct stages: initialization, input transformation, ensemble, and adversarial perturbation updating. Our techniques are designed to integrate seamlessly into these stages. For instance, we apply the random global momentum initialization and dual example in the initialization stage, high-frequency perturbation and spectral-based augmentations in the input transformation stage, gradient alignment, model shuffle, and asynchronous input transformation in the ensemble stage, and the scheduled step size in the adversarial perturbation updating stage. This structured approach ensures that each technique operates independently, contributing uniquely to the overall performance enhancement.
> > [**Experimental results**] To empirically validate the orthogonality of our techniques, we conducted experiments combining them in pairs. The studied tricks include random global momentum initialization (RGI), dual example with scheduled step size (DS), high-frequency spectral transformation (HT), gradient alignment (GA), asynchronous input transformation (AIT), and model shuffle (MS). We use the MI-FGSM as the optimization method for attack under the ensemble settings in our paper. We report the average attack success rates against the defense models for study, including the AT, FAT, HGD, RS, and NRP. The results are presented as follows,
> >
> > Tab. 1 Average attack success rates (%) of the combination of  our combined tricks on the defense methods.
> >| Tricks |  RGI | DS| HT| AIT | MS|GA|
> >| -------- |  -------- | -------- | -------- | -------- | -------- | -------- |
> >| RGI     |  48.2     | 51.5     | 53.4   |49.3   | 48.5  | 48.7  |
> >| DS     |    --     | 49.3     |55.9    | 51.8  | 50.6  | 53.9  |
> >| HT     |     --    | --     | 50.2   | 57.0  | 53.4  |  51.8 |
> >| AIT     |   --    |  --    |   -- | 48.5  | 49.3  |  49.0 |
> >| MS     |     --    |   --   |  --  | --  |  47.6 |  48.5 |
> >|GA|     --   |   --   |  --  | --  | --  |48.1   |
> >
> >The results demonstrate that the combination of different techniques leads to a significant improvement in performance. For example, combining random global momentum initialization (RGI) with dual example and scheduled step size (DS) resulted in a higher success rate (51.5%) compared to using either technique alone (RGI: 48.2%, DS: 49.3%). On the one hand, the RGI guides the optimization towards the global optima. On the other hand, more near-sample gradients are utilized by the scheduled step size to improve the adversarial transferability. This evidence strongly supports our claim of orthogonality and synergistic effectiveness.
>
> > **A3**: Our proposed tricks can be integrated into various attacks. As you suggeted, we update the results of our revised paper with the  use of  these methods, including MIG [a], TGR [b], NAA [c], and RAP[d],  with or without our proposed tricks to generate adversarial examples under the ensemble setting, and attack the advanced defense models and Google's vision API. We generate  $1,000$ adversarial examples to attack defense models and $100$ adversarial examples to attack Google's vision API. The results are depicted as follows,
> >
> > Tab.1 Attack success rates (%) of our combined tricks and various attacks on the defense methods and Google’s vision API.
> >| Method | Tricks | AT   | FAT  | HFD   | RS   | NRP  | Google |
> >|--------|--------|------|------|-------|------|------|--------|
> >| MIG [a]    |  $\times$  | 72.1 | 38.6 | 69.5  | 24.6 | 65.3 | 66.0   |
> >|   MIG [a]  | $\checkmark$ | 91.3 | 52.1 | 88.5  | 42.4 | 81.6 | 79.0   |
> >| TGR [b]   |  $\times$ | 75.1 | 35.3 | 82.4  | 29.5 | 73.0 | 70.0   |
> >|   TGR [b]  |  $\checkmark$ | 97.3 | 61.5 | 95.4  | 57.6 | 91.3 | 100.0  |
> >| NAA [c]   |  $\times$ | 69.5 | 34.3 | 80.2  | 25.5 | 64.6 | 59.0   |
> >|   NAA [c]  |  $\checkmark$  | 80.2 | 61.3 | 85.7  | 35.9 | 73.1 | 71.0   |
> >| RAP [d]    |  $\times$  | 59.2 | 33.5 | 75.3  | 20.6 | 58.1 | 52.0   |
> >|  RAP [d]  |  $\checkmark$  | 65.3 | 36.9 | 79.2  | 24.8 | 65.7 | 62.0   |
> >
> > The results clearly illustrate that applying our techniques significantly improves the adversarial transferability of these recent methods. This updated analysis and the corresponding results have been added to Section 4 of our revised manuscript.

---

> ### Author Response · Authors · 2023-11-22
>
> Dear Reviewer A7nY,
>           We have submitted our response to your questions and revised our paper as you suggested. We sincerely appreciate your valuable feedback on improving the quality of our paper.
>     Are there any additional questions or concerns we can answer?  Thanks for your reply!
>
> Sincerely,
> Authors

---

> > ### Comment · Reviewer_A7nY · 2023-11-23
> > **Thank you for your response**
> >
> > Most of my concerns have been addressed. I have raised my score to 6.

---

> > > ### Author Response · Authors · 2023-11-23
> > >
> > > Thanks for your positive comments and raising the score!
> > > We really appreciate your suggestions on improving our paper by providing the orthogonal study and comparison with novel methods.    We will polish our paper more clearly in the final revision.

---

### Official Review · Reviewer_Pj5y · 2023-10-30

**Soundness:** 3 good
**Presentation:** 3 good
**Contribution:** 2 fair
**Rating:** 5
**Confidence:** 4

**Summary:**

This paper proposes a bag of tricks to boost the transferability of adversarial examples, including techniques for gradient-based attacks, input transformation-based attacks, and ensemble attacks. The proposed tricks are evaluated extensively on ImageNet, against defense methods and Google Cloud Vision. Combining tricks boosts success rates, highlighting their complementary nature.

**Strengths:**

The paper tackles an important problem - improving adversarial transferability. The bag of tricks enhances attacks without architectural changes. The tricks are intuitive and easy to implement, requiring only minor modifications to existing methods.

**Weaknesses:**

1. For each model and attack, the ideal hyperparameters, including the iteration count and the scheduled step size, require fine-tuning, and the study lacks of theoretical direction for this.
2. While the tricks enhance transferability, the computational overhead and training time increase from additional steps like random initializations.
3. Effects of tricks diminish with certified defenses. More analysis is needed on certified robustness.

**Questions:**

See the above.

---

> ### Author Response · Authors · 2023-11-20
>
> ### Responses to Weakness & Questions
> > **A1**: Thank you for your insightful question. The theoretical analysis in understanding adversarial transferability remains less explored but is fundamentally important [1]. To the best of our knowledge, theoretical work on the effectiveness of attack methods in adversarial transferability is quite limited and challenging [2-5].
> >
> >  Our work is an empirical study, which starts from the hyper-parameter study, deriving lots of intuitive insights from these results and correspondingly proposing a bag of tricks to boost the adversarial transferability.  In particular, the numeric results in the hyper-parameter study suggest that adversarial transferability is much more sensitive to some basic settings than we thought. **We hope our empirical findings with extensive practical experiments can motivate theoretical insights and analysis on this sensitivity to understand adversarial transferability better.** We add more in-depth analysis in the hyper-parameter study of Section 3 and provide a detailed discussion, including the potential theoretical direction to utilize the insights, in Appendix G of our revision.
>
> > [1] Bose, Joey, et al. "Adversarial example games." NeurIPS 2020.
> > [2] Wang, Yilin, and Farzan Farnia. "On the role of generalization in transferability of adversarial examples." UAI 2023.
> > [3] Yang, Zhuolin, et al. "Trs: Transferability reduced ensemble via promoting gradient diversity and model smoothness." NeurIPS 2021.
> > [4] Zhang, Yechao, et al. "Why Does Little Robustness Help? A Further Step Towards Understanding Adversarial Transferability." S&P 2024.
> > [5] Zhao, Anqi, et al. "Minimizing Maximum Model Discrepancy for Transferable Black-box Targeted Attacks." CVPR 2023.
>
> > **A2**: **The computation overheads in adversarial attacks mainly derive from the forward and backward propagations.** Random global momentum initialization (RGI) with additional iterations definitely increases the computation overhead and training time (around $1.5 \times$). However, it is noteworthy that this increase in computation facilitates a substantial enhancement in adversarial transferability, with improvements of up to 9.2% in MI-FGSM. Given the significant enhancement in attack capability, the increased computational time is justifiable, especially from the perspective of an attacker prioritizing attack efficiency.
> >
> > Besides, most of our tricks do not take additional forward and backward propagations, which do not decay the computational efficiency. For instance, in our proposed tricks under the non-ensemble setting, the scheduled step size does not incur any additional computational cost. Additionally, the dual example and high-frequency transformation introduce a minor computational overhead with minimal impact on the overall computational duration.
> >
> > To support our argument, we conducted an experiment using the MI-FGSM method with VGG-16 to craft 10,000 adversarial images, integrating several tricks, including RGI. The results are as follows:
> >
> >| Method | MI |MI+RGI | MI+Scheduled step size | MI+Dual Example |MI+High-Frequency Transformation |
> >| -------- | -------- | -------- | -------- | -------- | -------- |
> >| Time (s) |    332   |   495       |  332        |   332       | 339 |
> >
> > It can be seen that the introduction of RGI results in around a 50% increase in computation time compared to the standard MI-FGSM, while others merely influence the efficiency. Considering the efficiency of adversarial example generation and the attack performance improvement, we think this additional time is acceptable for better performance in practical applications.
>
> > **A3**: **Due to the strong capacity of certified defense, most existing adversarial attacks diminish the performance, while the integration of our tricks can boost the attack performance.** As shown in Tab. 5 of our revised paper, while RS effectively defends the various advanced attack methods, the integration of our tricks to these methods consistently brings a significant improvement in attack success rates (17.9\% on average). It indicates that our tricks can be employed to attack more invariant robust features, thus improving the attack performance.

---

> ### Author Response · Authors · 2023-11-22
>
> Dear Reviewer Pj5y,
>   We have submitted our response to your questions and revised our paper as you suggested. We sincerely appreciate your valuable feedback on improving the quality of our paper.
>     Are there any additional questions or concerns we can answer?  Thanks for your reply!
>
> Sincerely,
> Authors

---

### Official Review · Reviewer_7z2g · 2023-10-31

**Soundness:** 4 excellent
**Presentation:** 4 excellent
**Contribution:** 4 excellent
**Rating:** 8
**Confidence:** 3

**Summary:**

The authors propose a comprehensive set of tricks to boost the adversarial transferability of existing transfer-based black-box attack methods. These tricks are tailored to different categories of attacks, including iterative gradient-based attacks (e.g., global momentum initialization,
scheduled step size, and dual example), input transformation-based attacks (e.g., spectral-based input transformation), and model-related ensemble attacks (e.g., gradient alignment, asynchronous input transformation, and model shuffle). An extensive study on tweaking the
fundamental but seemingly trivial hyper-parameters, such as the number of iterations, step size, and momentum coefficient, is also carried out. The proposed bag of tricks provides valuable insights into influencing factors of adversarial transferability and techniques for enhancing it.
Relevant research directions, such as developing novel and more robust defense methods for DNNs, could be inspired by this paper’s findings.

**Strengths:**

1. [Originality]:
• This is a valuable work that integrates and analyzes multiple factors contributing to adversarial transferability.
2. [Quality]:
• Experiments are extensive for validating the effectiveness and generality of their proposed tricks; for example, multiple surrogate models are used, including VGG-16, ResNet-18, DenseNet-121, and ViT.
• The tricks are proposed with a decent logical flow: the former trick’s weaknesses become the later trick’s motivation and hypothesis to verify.
3. [Clarity]:
• Detailed experiment settings and algorithm descriptions are provided in the Appendix.
• At the end of each subsection of an empirical study, there’s a “Takeaways” section that summarizes the authors’ observations and insights.
• At the end of each subsection of a proposed trick, there’s a “Results” section that analyzes the attack success rate (ASR) results, concludes the functionality and mechanism of the specific trick, and validates the author’s hypothesis if there’s one. Important key sentences are also highlighted in italic fonts.
• At the outset of each subsection of a proposed trick, the motivation and reasoning behind the proposal, as well as the authors’ hypothesis, are provided.
• Abundant plots, charts, and tables are provided to display the results across multiple methods and tricks, which is convenient and easy for readers to compare.
• Ablation study of various combinations of tricks is provided.
4. [Significance]:
• According to Table 5, the authors’ combined tricks yield the highest ASR across multiple defense methods and even 100% ASR on Google Vision API. Particularly, comparing their approach to the runner-up method, there’s a remarkable 18.1% improvement against the most powerful defense method – Random Smoothing (RS). This simulates a new line of research regarding effective defense design against attacks with high adversarial transferability.

**Weaknesses:**

Some minor typos:
• On page 3, at the beginning of Section 3, you stated: “We use eight surrogate/victim models, comprising...”. There should be seven only: 5 CNNs – VGG-16, ResNet-18, ResNet-101, DenseNet-121, and MobileNet; 2 transformers – ViT and Swin.
• On page 7, on the second last line of the Hyper-parameter study section, “achieves” should be “achieved”.
2. You provided explanations and insights or viewpoints for most of the observations you made from the experimental results but not each. Although some might be obvious, it’s still better that you provide a sentence or two explaining each observation. Specifically, for example:

• In Fig. 2(a), why GIMI-FGSM peak at T=5, which differs greatly from other gradient- based attack methods?
• In Fig. 2(b), why excessively large step sizes can harm attack performance?
• In Fig. 2(c), why does the performance degradation occur when the momentum decay factor is less than one, indicating momentum increase?
3. It would be better if results on speech applications could also be provided.
4. It would be better if recommended future defense directions could be provided.

**Questions:**

1. In Fig. 1, which tricks exactly are being used for the integrated tricks (the orange bar)?
2. You mentioned: “we choose 10,000 images from the ImageNet-1K dataset as our evaluation set.” How the 10,000 images were chosen? You also didn’t mention which
dataset you used to generate adversarial examples. Is it also ImageNet-1K?
3. For ensemble strategies, was there a specific reason why you used MI-FGSM attack to create adversarial perturbations? Would it be better if you provided more results with other attack methods to even strongly validate the effectiveness of ensemble strategies?

---

> ### Author Response · Authors · 2023-11-20
>
> ### Responses to Weaknesses
> > **A1**: Thanks for your careful review! We fixed these typo issues in our revision and polished the whole paper carefully.
>
> > **A2**: Thanks for your suggestion! we give a more in-depth analysis in the hyper-parameter study of our revision.
>
> > **A2.1**:   Different from other gradient-based methods, GIMI-FGSM initializes the momentum with a pre-computed global momentum. We infer that it is the global momentum that helps GIMI-FGSM skip the local optima near benign samples faster than others, but it is still easily influenced by multiple local optima away from the benign samples.    Thus, GIMI-FGSM has a peak performance at around $T=5$ in our experiment, which is much earlier than others.
> >
> > **A2.2**: There are two reasons for the performance degradation with excessively large step sizes. First, **it is easy to skip over the optima using the excessively large step size for optimization**, leading to poor adversarial transferability. Second, following the general setup of previous work, the crafted adversarial perturbation is limited within $\frac{16}{255}$. With the excessively large step size, **the generated perturbation easily exceeds the limit**, leading to being clipped to satisfy the constraint. This will fail the effectiveness of optimization in the current iteration, leading to performance degradation.
> >
> > **A2.3**: **When the momentum decay factor is less than one, it indicates a decreasing impact of historical gradients in the computation of momentum**, instead of the decreasement of momentum. Following previous studies [1,2,3], we apply the decay factor to the historical accumulated gradients, i.e.,
> >
> > $$m_{t+1}=\gamma \cdot m_{t} + g_{t+1},$$
> >
> > where we denote $m_{t}$ and $m_{t+1}$ as the momentum, $g_{t+1}$ as the novel gradient, and **$\gamma$ as the decay factor.** The same presentation is also consistently adopted in our algorithms in the appendix, such as Alg.1 (random global momentum initialization) in Appendix B.1 and Alg. 2 (dual example with ensemble strategy) in Appendix B.2. With decreasing $\gamma$, it is not good for stabilizing the optimization,  leading the optimization of adversarial examples easily **stuck in local optima** for performance degradation.
> >
> > Thanks for your point.  We revised our paper to make it more clear.
> > [1] Wang et al. Enhancing the transferability of adversarial attacks through variance tuning. CVPR 2021.
> > [2] Gao et al. Patch-wise attack for fooling deep neural network. ECCV 2020.
> > [3] Lin et al. Nesterov Accelerated Gradient and Scale Invariance for Adversarial Attacks. ICLR 2020.
>
> > **A3**: Following previous work [1], we conduct an experiment on attacking audio-visual models in the event classification task. For the surrogate model, we use the model that uses ResNet-18 for both audio and visual encoders and sum operation as the fusion module for features of audio and visual modalities. The targeted model involves two models, which respectively use VGG-16 and ResNet-34 as the visual and audio encoders. We use the audio-visual model with the ResNet-18 as the backbone to craft 200 adversarial examples and attack the other two models. The 200 audio-visual samples are randomly selected from the MIT-MUSIC dataset, which are classified correctly by the studied models. We use MI-FGSM to craft the adversarial examples and respectively apply the proposed random global initialization (RGI), scheduled step size, and dual example strategies to MI-FGSM. The results of attack success rates are reported as follows,
> >
> > | Audio-Visual Models | ResNet-18 | VGG-16 | ResNet-34|
> >| -------- | -------- | -------- | -------- |
> >|   MI   |  100.0    |  39.5    | 40.6  |
> >|MI+RGI   |   100.0  |   43.7   | 42.8 |
> >|MI+Scheduled step size   | 100.0     | 42.1     | 41.9 |
> >|MI+Dual example   |   100.0  | 45.2     | 46.9 |
> >
> > From the results, we can see a clear improvement (3.8% on average) by integrating the tricks into MI-FGSM. It shows the scalability of our tricks in audio-visual applications.  We add these results in Appendix H of our revision.
> >
> >[1] Tian, Yapeng, and Chenliang Xu. "Can audio-visual integration strengthen robustness under multimodal attacks?." CVPR 2021.
>
> > **A4**:  In the long run, future studies are expected to design novel training methods to improve the model robustness, e.g., adversarial training and certified defense by random smoothing. For real-world applications, it is valuable to explore input transformation-based methods to purify the adversarial perturbation, thus achieving a good balance between efficiency and robustness.

---

> > ### Comment · Reviewer_7z2g · 2023-11-22
> > **Thanks for the detailed author response.**
> >
> > Thank you for providing the response and revised manuscript. Overall, my concerns have been addressed. What I like about the paper is its attack success rate against adversarial training and other defenses. While I still insist, a more detailed investigation of the defense side may be necessary. For example, cross-attack adversarial training and attack, mixed L-norm attack training, and so on. Finally, I will stick to my decision to accept the paper.

---

> > > ### Author Response · Authors · 2023-11-22
> > >
> > > Dear Reviewer 7z2g,
> > >       Thanks for your help on improving the quality of our paper! We totally agree that it is really important to design advanced defense methods to improve the model robustness. We hope our study can be used to benchmark the robustness performance to evaluate the defense performance in the future.
> > > Thanks!
> > > Authors

---

> ### Author Response · Authors · 2023-11-20
>
> ### Responses to Questions
> > **A1**: In Fig. 1, we combine 5 tricks into different attacks, namely the random global momentum initialization, scheduled step size, dual example, high-frequency transformation, and spectral input transformations.
>
> > **A2**: For all experiments in our paper, we use the ImageNet-1K dataset. We randomly select $10$ images from each class in ImageNet-1K for evaluation. All of these images are classified correctly by $8$ studied models in our paper.
>
> > **A3**: MI-FGSM is generally adopted as the backbone attack method for various attack methods in many previous studies, such as DIM, TIM, Admix, etc. Thus, we also use the MI-FGSM as the base method under the ensemble setting to generate adversarial examples.
> >
> > We have provided the experimental results of applying our tricks to advanced ensemble-based attacks (SVRE and SSA) under the ensemble setting in Tab. D.3 of Appendix D.3.
> >
> > Here, to further verify the effectiveness of our proposed tricks, we replace MI-FGSM with two other gradient-based attacks, namely PI- and VMI-FGSM, under the same ensemble setting in our paper for evaluation. The results are depicted as follows,
> >
> >
> >
> >| Strategy | Trick | ASR (PI/VMI) |
> >| -------- | -------- | -------- |
> >| Loss     | --     |    75.3/77.2  |
> >| Loss     | GA     |  76.5/79.3    |
> >| Loss     | AIT     |   75.8/78.4   |
> >| Logit     | --     |  86.3/90.1    |
> >| Logit     | AIT     |   88.2/93.5   |
> >| Logit     | GA     |  88.1/91.7    |
> >| Prediction     | --     |   75.6/78.1   |
> >| Prediction     | GA     |  76.9/79.3    |
> >| Prediction     | AIT     |  78.5/81.6    |
> >| Longitude     | --     | 87.2/92.3     |
> >| Longitude     | MS     |  90.0/94.1    |
> >
> >From the results, we can see the integration of our tricks (GA, AIT, and MS) enhances the ensemble attack performance of PI- and VMI-FGSM, consistent with the results on MI-FGSM in our paper. It validates the effectiveness of ensemble strategies.

---

### Official Review · Reviewer_uKVA · 2023-11-01

**Soundness:** 3 good
**Presentation:** 3 good
**Contribution:** 3 good
**Rating:** 6
**Confidence:** 4

**Summary:**

This paper investigates transfer-based adversarial attacks and proposes a series of simple yet effective tricks to enhance the adversarial transferability of different category of attack methods. The authors conduct extensive experiments to demonstrate how individual trick can help crafting adversarial example and combining them together can significantly improve the attack performance on various defense models and Google vision API.

**Strengths:**

It is good empirical study work with detailed experiments.

**Weaknesses:**

- Although the tricks are simple and the technical novelty is limited, the authors do provide many insights into the limitations of existing methods. The limitations and corresponding solutions are well explained and hence easy to understand and follow.

- The experiments are thorough and well conducted overall, and a lot of statistical details are well presented regarding the comparison studies.

- In section 3.1 hyper-parameters study, the authors choose seven FGSM based approaches for the comparison study, however for tricks described in section 3.2, 3.3 & 3.4, only five methods are selected as shown in Table 1, 2, & 3. Why not keep in consistent in all the comparison study for gradient based attacks?

- For ensemble based attack, a recent study [1] is not compared. I am wondering if any of the tricks applies as well.

- For input transformation methods in related works, previous study [2] is not properly discussed as well.

### References
[1] Chen, H., Zhang, Y., Dong, Y., & Zhu, J. (2023). Rethinking Model Ensemble in Transfer-based Adversarial Attacks. arXiv preprint arXiv:2303.09105

[2] Dong, Y., Pang, T., Su, H., & Zhu, J. (2019). Evading defenses to transferable adversarial examples by translation-invariant attacks. In Proceedings of the IEEE/CVF Conference on Computer Vision and Pattern Recognition (pp. 4312-4321)

**Questions:**

See weakness above

---

> ### Author Response · Authors · 2023-11-20
> **Responses to Weakness & Questions 1-3**
>
> ### Responses to Weaknesses and Questions
> >**A.1 && A.2**: We thank the reviewer for reading our paper carefully and giving the above positive comments！
>
> > **A.3**: In the hyper-parameters study, we select 7 FGSM-based approaches, including I-FGSM, MI-FGSM, NI-FGSM, PI-FGSM, EMI-FGSM, VMI-FGSM, GIMI-FGSM, for a comprehensive comparison.
> >
> > In Section 3.2, we study the influence of **momentum initialization** and the proposed trick, random global momentum initialization (RGI). We did not include the performance comparison with I-FGSM because **I-FGSM does not use the momentum** for crafting adversarial perturbation. **We already include the comparison with GIMI-FGSM in Tab. 1**. We denote the global momentum initialization technique as GI in Tab. 1 and apply it to compared methods, including MI-, NI-, PI-, EMI-, and VMI-FGSM. **The integration of GI- into MI-FGSM is the GIMI-FGSM.** Thus, we actually compare 6 methods in this section. We will clarify it in our revision.
> >
> > In Sections 3.3 and 3.4, we study the performance of proposed tricks, namely the scheduled step size and dual example.  **The proposed tricks aim sample better gradients around the input samples** for crafting transferable adversarial examples.  In the current version, we do not include the **NI-FGSM** and **EMI-FGSM** because they adopt **look-ahead** (calculating the gradient on a sample around the input sample) for gradient computation to skip the local optima, while **the other five methods** focus on **using gradients better to update directly** the adversarial example, which are more matched with our proposed tricks. As you suggested, Following the same setting in our paper, we also integrate our tricks into NI-FGSM and EMI-FGSM, as presented in the following tables.
> >
> > Tab. 1: Average attack success rates (%) of gradient-based attacks using various scheduled step sizes, including NI-FGSM and EMI-FGSM. We report the performance under the default setting (T = 10)/optimal setting. We denote ”Ori.” as the identity step size. We use the VGG-16 as the surrogate model and evaluate the adversarial transferability on other 7 models.
> >
> > | Step     | NI-FGSM  		| EMI-FGSM 		   |
> > | -------- | --------|--------  |
> > | Ori.     | 57.4/58.6		| 65.0/67.1		   |
> > | log      | 58.6/59.1		| 65.9/67.4		   |
> > | linear   | **60.4**/**60.7**| **66.7**/**68.3**|
> > | exp      | 57.9/57.9		| 63.8/64.5		   |
> > | pvalue   | 58.3/59.0		| 64.2/64.9		   |
> >
> > Tab. 2 : Average attack success rates (%) when applying dual example w/o or w ensemble strategy using various sequences as step size to NI-FGSM and EMI-FGSM. We denote ”N/A” as vanilla adversarial attacks, and others as dual example with scheduled step size. We use the VGG-16 as the surrogate model and evaluate the adversarial transferability on the other 7 models.
> > | Step     | NI-FGSM  	| EMI-FGSM |
> > | -------- | -------- | -------- |
> > | N.A.     |  57.4    	|    65.0  |
> > | Ori.     |59.3/64.0 	|66.3/68.9 |
> > | log      |**62.5**/66.9 |63.8/**69.4** |
> > | linear   |61.7/**67.3** |**67.1**/**69.4** |
> > | exp      |61.3/65.8 	|65.9/67.3 |
> > | pvalue   |62.0/64.2 	|66.2/68.7 |
> >
> > In Tab.1, we present the performance of the application of scheduled step size. We can see a clear  attack success rate improvement by simply adopting a suitable scheduled step size sequence. With the linear step size, the NI- and EMI-FGSM have an improvement of $2.6\%$ and $1.5\%$ on average, respectively.
> > In Tab.2, we present the performance of the application of dual example with scheduled step size. We can see a significant improvement from the numeric results. Specifically, by using ensemble strategy of the dual example, NI- and EMI-FGSM have a large improvement of up to $9.9\%$ and $4.4\%$ in average attack success rate.
> > On the one hand, we can observe that the proposed tricks are also applicable to these attacks, which further validate their superiority and generality. On the other hand, consistent with our paper, different attack methods have different suitable scheduled step sizes for the best performance. It remains an open problem about how to decide the best scheduled or dynamic step size to boost the adversarial transferability. We hope our results can motivate future work researching this interesting problem.
> > **We added the results to Tab.2 and Tab.3 of our revision as you suggested.** Thanks.

---

> ### Author Response · Authors · 2023-11-20
> **Responses to Questions 4-5**
>
> ### Responses to Weaknesses and Questions
> > **A.4**: To verify the scalability of our proposed tricks, we respectively integrate the random global momentum initialization (RGI), dual example (DE), gradient alignment (GA), asynchronous input transformation (AIT), and model shuffle (MS) into the momentum- and ensemble-based method (MI-CWA) [2]. Following the same setting in our paper, we evaluate the attack performance against the defense methods. The results are depicted in the following table.
> >
> > | Method     | AT | FAT| HGD| RS| NRP|
> > |--------|--------|--------|--------|--------|--------|
> > | MI-CWA      |  82.7  |    45.2  | 89.1| 39.2| 83.5|
> > | MI-CWA+RGI      |  83.1  |    45.7  | 90.5| 43.0| 84.6|
> > | MI-CWA+DE      |  86.9  |    48.5  | 91.6| 44.7| 85.2|
> > | MI-CWA+GA      | 82.8  |    45.2  | 89.0| 39.4| 83.7|
> > | MI-CWA+AIT      |  92.8  |    55.4  | 97.2| 51.5| 87.2|
> > | MI-CWA+MS      |  83.2  |    45.5  | 89.1| 40.0| 83.9|
> > | MI-CWA+combination | 99.1  |    62.4  | 100.0| 59.1| 92.8|
> >
> > From the results, we can see the introduction of different tricks brings different improvements to the attack performance against various models. The  MI-CWA integrated with combinational tricks achieved the best performance, surpassing the best result achieved by VMI+comnination in our paper with an improvement of 2.33\% on average.   Although MI-CWA achieves the SOTA performance among the ensemble-based methods, our tricks can still boost the adversarial transferability, which further validates the generality and superiority of the proposed tricks.
> > For a better demonstration of the effectiveness and scalability of our tricks to advanced attacks, we added it to our discussion and the corresponding results in Tab.5 of our revision.
> >
> > [1] Chen, H., Zhang, Y., Dong, Y., & Zhu, J. (2023). Rethinking Model Ensemble in Transfer-based Adversarial Attacks. arXiv preprint arXiv:2303.09105
>
> > **A.5**: TIM [2] adopts Gaussian smooth on the gradient to approximate the average gradient of a set of translated images to update the adversary, which is a classical input transformation-based attack method. In our paper, however, we mainly study the effect of number of copies and diversity on the performance of input transformation-based attack methods. Thus, we do not include the comparison with TIM in our main paper. We have the comparison with TIM in our supplementary, shown in Tab. F9. We discussed it in Section 2 of the revision as you suggested. Thanks.
> >
> > [2] Dong, Yinpeng, et al. "Evading defenses to transferable adversarial examples by translation-invariant attacks." CVPR 2019.

---

> ### Author Response · Authors · 2023-11-22
>
> Dear Reviewer uKVA,
>          We have submitted our response to your questions and revised our paper as you suggested. We sincerely appreciate your valuable feedback on improving the quality of our paper.
>     Are there any additional questions or concerns we can answer?  Thanks for your reply!
>
> Sincerely,
> Authors

---

> > ### Comment · Reviewer_uKVA · 2023-11-23
> >
> > Thanks the authors to provide detailed response. I have no further questions.

---

> > > ### Author Response · Authors · 2023-11-23
> > >
> > > Thanks for your reply and positive comments. We appreciate your suggestion on improving our paper by providing consistent evaluation results.

---

### Author Response · Authors · 2023-11-23

## Summary of Changes
We thank all the reviewers for their insightful suggestions on improving the quality of our paper.  Approaching the deadline of the discussion period, we would like to  summarize the changes to our paper as follows,
1.  [**In-depth analysis**] We add more analysis about the hyper-parameter study in Section 3.
2.  [**Paper consistency**] We add NI- and EMI-FGSM in Tab. 2 and Tab. 3 to keep it consistent in all comparison studies for gradient-based attacks.
3.  [**Comparison with recent works**] We add more comparisons with recent works in Tab.5 to demonstrate better the scalability and effectiveness of our proposed tricks in boosting the adversarial transferability.
4.  [**Comprehensive discussion**] We add the orthogonal study of our tricks in Appendix C.5, provide more discussion on the hyper-parameter study in Appendix G, and present the multi-modal applications using our proposed tricks for a more comprehensive study.

We have highlighted the changes in our paper in blue color. Thanks for your review and suggestions again! We are open to answering any additional concerns and questions!

---

### Meta-Review · Area_Chair_ibYi · 2023-12-16

**Metareview:**

Adversarial examples generated by whitebox surrogate models often do not transfer to blackbox victim models. This paper provides detailed experiments to evaluate effects of different settings on the transfer attack success rate. Parameters under study include number of  iterations, step size and schedule, momentum initialization, input transformations, and ensemble strategies. Most of the experiments are performed with imagenet dataset for untargeted attacks on classifiers.

Strengths:
+ It is good empirical study work with detailed experiments.
+ The paper is well written and the main takeaways and insights from the experiments are clearly presented.

+/- The paper does not introduce any novel idea and primarily evaluates performance of existing methods in different settings.

Weaknesses:
- The paper mainly focuses on untargeted attacks, which are relatively easy compared to targeted attacks. This is especially true with perturbation budget of 16/255.
- The effect of perturbation budget is not studied in the paper.
- Selection of surrogates and victims makes a difference. This aspect is not studied in detail in the paper.
- The paper has limited experiments on robustly-trained models
- Ensemble-based methods did not test strong attacks (e.g., Liu et al DELVING INTO TRANSFERABLE ADVERSARIAL EXAMPLES AND BLACK-BOX ATTACKS and follow up ensemble methods report almost 90% success rate on untargeted attacks)
- The paper did not compare the bag of tricks against other robust attack methods. For instance, Auto Attack (https://github.com/fra31/auto-attack) is considered state-of-the-art method for testing adversarial attacks and defense methods. It is not clear what advantage this paper provides over existing methods such as auto attack.
- The paper did not discuss the effects of the adversarial loss functions (e.g., most of the experiments use entropy maximization; C&W loss is sometimes a better loss function)

**Justification For Why Not Higher Score:**

- A fair comparison with existing robust attack methods, such as Auto Attack or similar benchmarks, could be helpful.
- A fair comparison with existing ensemble methods that achieve nearly 90% attack success rate on untargeted attacks would be helpful
- Effect of the proposed bag of tricks on targeted attacks would be helpful
- Experiments with robust classifiers (with defense) would be helpful for this paper and broader research community.

**Justification For Why Not Lower Score:**

N/A

---

### Decision · Program_Chairs · 2024-01-16

Reject